# Epistatic selection on a selfish *Segregation Distorter* supergene – drive, recombination, and genetic load

**Beatriz Navarro-Dominguez[1†], Ching-Ho Chang[1‡], Cara L Brand[1§], Christina A Muirhead[1,2], Daven C Presgraves[1]\*, Amanda M Larracuente[1]\***

[1]Department of Biology, University of Rochester, Rochester, United States; [2]Ronin Institute, Montclair, United States

**Abstract** Meiotic drive supergenes are complexes of alleles at linked loci that together subvert Mendelian segregation resulting in preferential transmission. In males, the most common mechanism of drive involves the disruption of sperm bearing one of a pair of alternative alleles. While at least two loci are important for male drive—the driver and the target—linked modifiers can enhance drive, creating selection pressure to suppress recombination. In this work, we investigate the evolution and genomic consequences of an autosomal, multilocus, male meiotic drive system, *Segregation Distorter* (*SD*) in the fruit fly, *Drosophila melanogaster*. In African populations, the predominant *SD* chromosome variant, *SD-Mal*, is characterized by two overlapping, paracentric inversions on chromosome arm *2R* and nearly perfect (~100%) transmission. We study the *SD-Mal* system in detail, exploring its components, chromosomal structure, and evolutionary history. Our findings reveal a recent chromosome-scale selective sweep mediated by strong epistatic selection for haplotypes carrying *Sd*, the main driving allele, and one or more factors within the double inversion. While most *SD-Mal* chromosomes are homozygous lethal, *SD-Mal* haplotypes can recombine with other, complementing haplotypes via crossing over, and with wildtype chromosomes via gene conversion. *SD-Mal* chromosomes have nevertheless accumulated lethal mutations, excess non-synonymous mutations, and excess transposable element insertions. Therefore, *SD-Mal* haplotypes evolve as a small, semi-isolated subpopulation with a history of strong selection. These results may explain the evolutionary turnover of *SD* haplotypes in different populations around the world and have implications for supergene evolution broadly.

**\*For correspondence:**
daven.presgraves@rochester.edu (DCP);
alarracu@UR.Rochester.edu (AML)

**Present address:** [†]Departamento de Genética, Universidad de Granada, Granada, Spain; [‡]Fred Hutchinson Cancer Research Center, Seattle, United States; [§]Department of Biology, University of Pennsylvania, Philadelphia, United States

**Competing interest:** The authors declare that no competing interests exist.

## Editor's evaluation

The work advances our understanding of the *Segregation Distorter* (*SD*) complex in *Drosophila melanogaster*. *SD*, the classic example of a selfish chromosome, consists of two tightly linked genetic elements and thus qualifies as a supergene. The work also excels through particularly careful analyses.

## Introduction

Supergenes are clusters of linked loci that control variation in complex phenotypes. Some supergenes mediate adaptive polymorphisms that are maintained by some form of frequency- or density-dependent natural selection, as in, for example, mimicry in butterflies, self-incompatibility in plants, plumage polymorphisms in birds, and heteromorphic sex chromosomes (see *Schwander et al., 2014*; *Thompson and Jiggins, 2014*, for review). Other supergenes are maintained by selfish social behaviors that enhance the fitness of carriers at the expense of non-carriers, as in some ant species (*Keller*

and Ross, 1998; Wang et al., 2013). Still other supergenes are maintained by their ability to achieve selfish, better-than-Mendelian transmission during gametogenesis, as in the so-called meiotic drive complexes found in fungi, insects, and mammals (Lyon, 2003; Larracuente and Presgraves, 2012; Lindholm et al., 2016; Svedberg et al., 2018; Fuller et al., 2020).

Meiotic drive complexes gain transmission advantages at the expense of other loci and their hosts. In heterozygous carriers of male drive complexes in animals, the driver disables spermatids that bear drive-sensitive target alleles (Larracuente and Presgraves, 2012; Lindholm et al., 2016). To spread in the population, the driver must be linked in a *cis*-arrangement to a drive-resistant (insensitive) target allele (Charlesworth and Hartl, 1978). Recombination between the driver and target can result in a 'suicide' haplotype that distorts against itself (Sandler and Carpenter, 1972; Hartl, 1974). These epistatic interactions between driver and target lead to selection for modifiers of recombination that tighten linkage, such as chromosomal inversions (Charlesworth and Hartl, 1978; Schwander et al., 2014; Thompson and Jiggins, 2014; Charlesworth, 2016). Like most supergenes (Charlesworth and Charlesworth, 1975; Turner, 1977), meiotic drive complexes originate from two or more loci with some degree of initial linkage. Successful drivers thus tend to be located in regions of low recombination, such as non-recombining sex chromosomes (Hamilton, 1967; Hurst and Pomiankowski, 1991), centromeric regions, or in chromosomal inversions of autosomes (Lyon, 2003; Larracuente and Presgraves, 2012; Lindholm et al., 2016; Svedberg et al., 2018).

The short-term benefits of reduced recombination can entail long-term costs. Chromosomal inversions that lock supergene loci together can also incidentally capture linked loci, which causes large chromosomal regions to segregate as blocks. Due to reduced recombination, the efficacy of natural selection in these regions is compromised: deleterious mutations can accumulate, and beneficial ones are more readily lost (Muller, 1964; Hill and Robertson, 1968; Felsenstein, 1974; Charlesworth et al., 2009). Many meiotic drive complexes are thus homozygous lethal or sterile. The degeneration of drive haplotypes is not inevitable, however. Different drive haplotypes that complement one another may be able to recombine, if only among themselves (Dod et al., 2003; Presgraves et al., 2009; Brand et al., 2015). Gene conversion from wildtype chromosomes may also ameliorate the genetic load of supergenes (Uyenoyama, 2005; Wang et al., 2013; Tuttle et al., 2016; Branco et al., 2018; Stolle et al., 2019; Brelsford et al., 2020). Male meiotic drive complexes thus represent a class of selfish supergenes that evolve and persist via the interaction of drive, recombination, and natural selection.

Here, we focus on the evolutionary genetics of *Segregation Distorter* (*SD*), a well-known autosomal meiotic drive complex in *Drosophila melanogaster* (Sandler et al., 1959). In heterozygous males, *SD* disables sperm bearing drive-sensitive wildtype chromosomes via a chromatin condensation defect (Hartl et al., 1967; Temin et al., 1991). *SD* has two main components: the driver, *Segregation Distorter* (*Sd*), is a truncated duplication of the gene *RanGAP* located in chromosome arm *2L* (Powers and Ganetzky, 1991; Merrill et al., 1999; Kusano et al., 2001); and the target of drive, *Responder* (*Rsp*), is a block of satellite DNA in the pericentromeric heterochromatin of *2R*. Previous studies of *SD* chromosomes have detected linked upward modifiers of drive, including *Enhancer of SD* (*E[SD]*) on *2L* and several others on *2R* (Sandler and Hiraizumi, 1960; Miklos, 1972; Ganetzky, 1977; Hiraizumi et al., 1980; Brittnacher and Ganetzky, 1984), but their molecular identities are unknown. *Sd-RanGAP* and *Rsp* straddle the centromere, a region of reduced recombination, and some *SD* chromosomes bear pericentric inversions that presumably further tighten linkage among these loci. In heterozygotes with a pericentric inversion, recombination in the inverted region generates aneuploids and therefore reduced fertility, although this effect might be mitigated by strong suppression of recombination (Coyne et al., 1993). Many *SD* chromosomes also bear paracentric inversions on *2R* (reviewed in Lyttle, 1991; Larracuente and Presgraves, 2012). Although recombination between paracentric inversions and the main components of *SD* is possible, their strong association implies a role for epistatic selection in the evolution of these supergenes (Larracuente and Presgraves, 2012).

While *SD* is present at low population frequencies (<5%) around the world (Temin et al., 1991; Larracuente and Presgraves, 2012), *Sd-RanGAP* appears to have originated in sub-Saharan Africa, the ancestral geographic range of *D. melanogaster*, survived the out-of-Africa bottleneck, and spread to the rest of the world (Presgraves et al., 2009; Brand et al., 2015). Multiple factors likely contribute to the low frequency of *SD* in populations: negative selection, insensitive *Rsp* alleles, and unlinked suppressors (reviewed in Larracuente and Presgraves, 2012). Two independent longitudinal studies

suggest that *SD* haplotypes can replace each other in populations over short time scales (<30 years) (***Temin and Kreber, 1981***; ***Brand et al., 2015***) without major changes in the overall population frequency of *SD* (***Temin and Kreber, 1981***). The predominant *SD* variant in Africa is *SD-Mal*, which recently swept across the entire continent (***Presgraves et al., 2009***; ***Brand et al., 2015***). *SD-Mal* has a pair of rare, African-endemic, overlapping paracentric inversions spanning ~40% of *2R:In(2R)51B6–11;55E3–12* and *In(2R)44F3–12;54E3–10*, hereafter collectively referred to as *In(2R)Mal* (***Aulard et al., 2002***; ***Presgraves et al., 2009***). *SD-Mal* chromosomes are particularly strong drivers, with ~100% transmission. Notably, recombinant chromosomes bearing the *Sd-RanGAP* duplication from this haplotype but lacking the inversions do not drive (***Presgraves et al., 2009***), suggesting that *In(2R)Mal* is essential for *SD-Mal* drive. We therefore expect strong epistatic selection to enforce the association of *Sd-RanGAP* and *In(2R)Mal*. The functional role of *In(2R)Mal* for drive is still unclear: do these inversions function to suppress recombination between *Sd-RanGAP* and a major distal enhancer on *2R*, or do they contain a major enhancer?

Here, we combine genetic and population genomic approaches to study *SD-Mal* haplotypes sampled from a single population in Zambia, the putative ancestral range of *D. melanogaster* (***Pool et al., 2012***). We address four issues. First, we reveal the structural features of the *SD-Mal* haplotype, including the organization of the insensitive *Rsp* allele and the *In(2R)Mal* rearrangements. Second, we characterize the genetic function of *In(2R)Mal* and its role in drive. Third, we infer the population genetic history of the rapid rise in frequency of *SD-Mal* in Zambia. And fourth, we explore the evolutionary consequences of reduced recombination on *SD-Mal* haplotypes. Our results show that *SD-Mal* experienced a recent chromosome-scale selective sweep mediated by epistatic selection and has, as a consequence of its reduced population recombination rate, accumulated excess non-synonymous mutations and transposable element (TE) insertions. The *SD-Mal* haplotype is a supergene that evolves as a small, semi-isolated subpopulation in which complementing *SD-Mal* chromosomes can recombine *inter se* via crossing over and with wildtype chromosomes via gene conversion. These results have implications for supergene evolution and may explain the enigmatic evolutionary turnover of *SD* haplotypes in different populations around the world.

## Results and discussion

To investigate the evolutionary genomics of *SD-Mal*, we sequenced haploid embryos from nine driving *SD-Mal* haplotypes sampled from a single population in Zambia (***Brand et al., 2015***), the putative ancestral range of *D. melanogaster* (***Pool et al., 2012***). Illumina read depth among samples ranged between ~46 and 67× (***Supplementary file 1***; BioProject PRJNA649752 in NCBI). Additionally, we obtained ~12× coverage with long-read Nanopore sequencing of one homozygous viable line, *SD-ZI125*, to create a de novo assembly of a representative *SD-Mal* haplotype (BioProject PRJNA649752 in NCBI; assembly in ***Navarro-Dominguez et al., 2022a***). We use these data to study the evolution of *SD-Mal* structure, diversity, and recombination.

### Chromosomal features of the *SD-Mal* supergene

The *SD-Mal* haplotype has at least three key features: the main drive locus, the *Sd-RanGAP* duplication on *2L*; an insensitive *Responder* (*Rsp^i^*) in *2R* heterochromatin; and the paracentric *In(2R)Mal* arrangement on chromosome *2R* (***Figure 1***). We used our long-read and short-read sequence data for *SD-ZI125* to confirm the structure of the duplication (***Figure 1A***) and then validated features in the other *SD-Mal* haplotypes. All *SD-Mal* chromosomes have the *Sd-RanGAP* duplication at the same location as the parent gene on chromosome *2L* (see also ***Brand et al., 2015***). The *Rsp* locus, the target of *SD,* corresponds to a block of ~120 bp satellite repeats in *2R* heterochromatin (***Figure 1B***; ***Wu et al., 1988***). The reference genome, *Iso-1*, has a *Rsp^s^* allele corresponding to a primary *Rsp* locus containing two blocks of tandem *Rsp* repeats—*Rsp-proximal* and *Rsp-major*—with ~1000 copies of the *Rsp* satellite repeat interrupted by TEs (***Khost et al., 2017***). A small number of *Rsp* repeats exist outside of the primary *Rsp* locus, although they are not known to be targeted by *SD*. There are three of these additional *Rsp* loci in *Iso-1*: ~ 10 copies in *2R*, distal to the major *Rsp* locus (*Rsp-minor*); a single copy at the distal end of *2R* (60A); and ~12 copies in *3L* (***Houtchens and Lyttle, 2003***; ***Larracuente, 2014***; ***Khost et al., 2017***). The genomes of *SD* flies carry ~20 copies of *Rsp* (***Wu et al., 1988***; ***Pimpinelli and Dimitri, 1989***), but the organization of the primary *Rsp* locus on *SD* chromosomes is

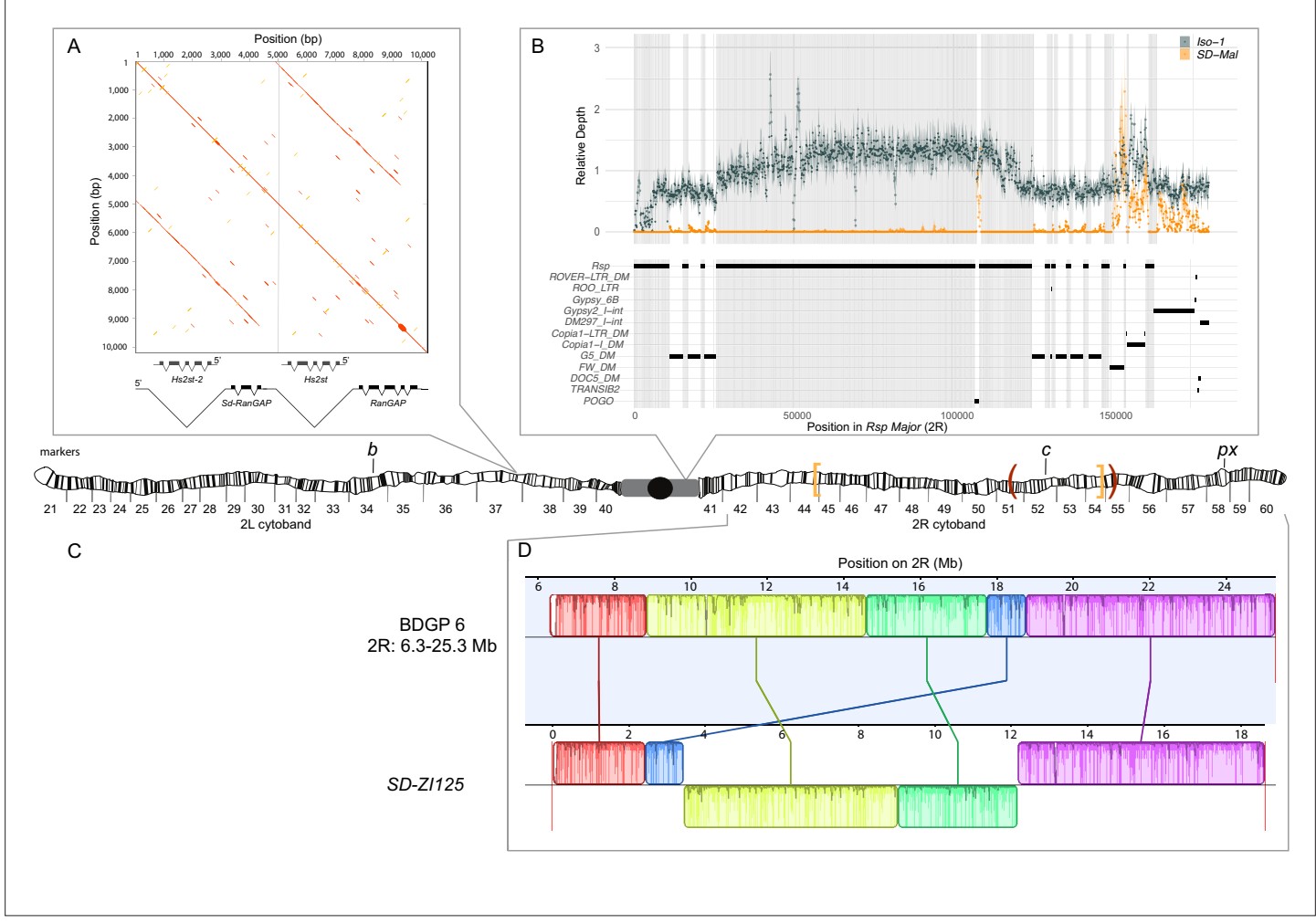

**Figure 1.** Map depicting the chromosomal features of the *SD-Mal* chromosome. The schematic shows the cytogenetic map of chromosomes *2L* and *2R* (redrawn based on images in *Lefevre, 1976*) and the major features of the chromosome. (**A**) Dotplot showing that the *Sd* locus is a partial duplication of the gene *RanGAP* (in black), located at band *37D2-6*. The gene *Hs2st* occurs in the first intron of *RanGAP*, and it is also duplicated in the *Sd* locus (*Hs2st-2*). (**B**) The *Rsp-major* locus is an array of tandem repeats located in the pericentric heterochromatin (band *h39*). Read mapping to a reference genome containing *2R* pericentric heterochromatin (*Iso1* strain, see *Chang and Larracuente, 2019*) shows that *SD-Mal* chromosomes do not have any *Rsp* repeats in the *Rsp-major* locus, consistent with being insensitive to distortion by *Sd* (*Rsp$^i$*) (orange, high relative coverage regions correspond to transposable element interspersed), in contrast with *Iso-1*, which is sensitive (*Rsp$^s$*). The tracks below indicate the presence of types of repetitive elements found at this locus. Black lines indicate the presence of a repeat type in the reference genome. Gray shading shows where *Rsp* repeats are in the reference genome. (**C**) Two paracentric, overlapping inversions constitute the *In(2R)Mal* arrangement shown on the schematic of polytene chromosomes: *In(2R)51BC;55E* (*In(2R)Mal-p*) in orange brackets and *In(2R)44F;54E* (*In(2R)Mal-d*) in red parentheses. Pericentromeric heterochromatin and the centromere are represented by a gray rectangle and black circle, respectively. (**D**) Our assembly based on long-read sequencing data provide the exact breakpoints of *In(2R)Mal* and confirms that the distal inversion (*Dmel.r6*, *2R*:14,591,034–18,774,475) occurred first, and the proximal inversion (*Dmel.r6*, *2R*:8,855,601–15,616,195) followed, overlapping ~1 Mb with the distal inversion. The colored rectangles correspond to locally collinear blocks of sequence with the height of lines within the block corresponding to average sequence conservation in the aligned region (*Darling et al., 2010*). Blocks below the center black line indicate regions that align in the reverse complement orientation. Vertical red lines indicate the end of the assembled chromosomes. Visible marker locations used for generating recombinants (*b* (34D1), *c* (52D1), and *px* (58E4-58E8)) are indicated on the cytogenetic map (*Lefevre, 1976*).

The online version of this article includes the following figure supplement(s) for figure 1:

**Figure supplement 1.** Estimated abundance of *Rsp* repeats at each *Rsp* locus in the reference *Iso-1* genome and *SD-Mal*.

**Figure supplement 2.** Possible rearrangements of synteny blocks to generate a double inversion.

**Figure supplement 3.** Model of the *In(2R)Mal* rearrangement.

**Figure supplement 4.** Crossing scheme to generate recombinants along *SD-Mal*.

**Table 1.** Frequency of recombinants along *SD-Mal* chromosomes vs. wildtype (*OreR*) chromosomes.

| Cross | N | n | $n_R$ (b–c) | $n_R$ (c–px) | d (b–c) | d (c–px) |
|---|---|---|---|---|---|---|
| *OreR/b c px x b c px/b c px* | 15 | 1,716 | 418 | 372 | 26.62 | 23.21 |
| *SD-Mal/b c px x b c px/b c px* | 11 | 1,820 | 211 | 32 | 11.81 | 1.76 |

*N*, number of crosses; *n*, total progeny scored; $n_R$, number of recombinants, *d*, genetic distance in cM, Kosambi-corrected.

unknown. To characterize the *Rsp* locus of the *SD-Mal* haplotype, we mapped *SD-Mal* reads to an *Iso-1* reference genome (see **Khost et al., 2017**). As expected, reads from *Iso-1* reference are distributed across the whole *Rsp-major* region. For *SD-Mal* chromosomes, however, very few reads map to the *Rsp* repeats at the *Rsp-major* (**Figure 1B**). This suggests that all *SD-Mal* have a complete deletion of the primary *Rsp* locus containing *Rsp-proximal* and *Rsp-major* and that the only *Rsp* copies in the *SD-Mal* genomes are the minor *Rsp* loci in chromosomes *2R* and *3L* (**Figure 1—figure supplement 1**).

The complex *In(2R)Mal* inversion is distal to the *Rsp* locus on chromosome *2R* (**Figure 1C**). We used our *SD-ZI125* assembly to determine the precise breakpoints of these inversions. Relative to the standard *D. melanogaster 2R* scaffold (BDGP6), *SD-ZI125* has three large, rearranged blocks of sequence corresponding to *In(2R)Mal* (**Figure 1C**): a 1.03 Mb block collinear with the reference but shifted proximally; a second inverted 5.74 Mb block; and a third inverted 3.16 Mb block. From this organization, we infer that the distal inversion, which we refer to as *In(2R)Mal-d*, occurred first and spanned 4.18 Mb (approx. *2R*:14,591,003–18,774,475). The proximal inversion, which we refer to as *In(2R)Mal-p*, occurred second and spanned 6.76 Mb, with 1.02 Mb overlapping with the proximal region of *In(2R)Mal-d* (approx. *2R*:8,855,602–17,749,310). Note that any rearrangement different than distal first, proximal second, leads to a different outcome (**Figure 1—figure supplement 2**). All four breakpoints of the *In(2R)Mal* rearrangement involve simple joins of unique sequence. Three of these four breakpoints span genes (**Figure 1—figure supplement 3**): *sns* (*2R*:8,798,489–8,856,091), *CG10931* (*2R*:17,748,935–17,750,136), and *Mctp* (*2R*:18,761,758–18,774,824). The CDSs of both *sns* and *Mctp* remain intact in the *In(2R)Mal* arrangement, with the inversion disrupting their 3' UTRs. Neither of these two genes is expressed in testes (https://flybase.org/reports/FBgn0024189; https://flybase.org/reports/FBgn0034389; **Chintapalli et al., 2007**; FB2021_06; **Larkin et al., 2021**), making it unlikely that they affect drive. *In(2R)Mal-p* disrupts the CDS of *CG10931*, which is a histone methyltransferase with high expression levels in testis (https://flybase.org/reports/FBgn0034274; **Chintapalli et al., 2007**; FB2021_06; **Larkin et al., 2021**). Even for genes that are not directly interrupted by the inversion breakpoints, the chromosomal rearrangements may disrupt the regulation of nearby genes if, for example, they affect the organization of topologically associating domains (TADs; reviewed in **Spielmann et al., 2018**). The *In(2R)Mal* inversion breakpoints disrupt physical domains reported in **Hou et al., 2012**, however inversion-mediated disruptions of TAD boundaries do not necessarily affect gene expression (**Ghavi-Helm et al., 2019**). Future work is required to determine if the inversions affect gene expression near the breakpoints and if *CG10931* has a role in the *SD-Mal* drive phenotype.

**Table 2.** Strength of segregation distortion in recombinants of *SD-ZI125*.

| | Genotype | Markers | N | n | ±SE | k | ±SE | k* | ±SE | p-value (k* = 0.5) |
|---|---|---|---|---|---|---|---|---|---|---|
| 1 | *Sd In(2R)Mal* | *+ + + and b + +* | 112 | 90.3 | 6.03 | 0.99 | 0.00 | 0.98 | 0.00 | <0.0001 |
| 2 | *Sd In(2R)Mal* | *+ + px* | 71 | 118.8 | 9.48 | 0.97 | 0.01 | 0.96 | 0.01 | <0.0001 |
| 3 | *Sd In(2R)Mal⁺* | *+ c px* | 19 | 147.6 | 14.39 | 0.54 | 0.01 | 0.51 | 0.01 | 0.3082 |
| 4 | *Sd⁺ In(2R)Mal* | *b + +* | 24 | 124.8 | 10.31 | 0.68 | 0.02 | 0.55 | 0.03 | 0.0572 |
| 5 | *Sd⁺ In(2R)Mal⁺* | *+ c px* | 65 | 120.4 | 8.32 | 0.53 | 0.01 | 0.51 | 0.01 | 0.3586 |

Chromosome *2* markers are *black* (*b*), *curved* (*c*), and *plexus* (*px*). *N*, number of crosses; *n*, average number of progeny from the crosses; *SE*, standard error; *k*, average proportion of progeny inheriting the recombinant $SD_r$ chromosome from $SD_r/b \, c \, px$ males; *k\**, average proportion of progeny inheriting the partial $SD_r$ chromosome from $SD_r/b \, c \, px$ males, corrected for viability. p-values reported by a single sample t-test with a null hypothesis of *k\** = 0.5, as expected for Mendelian segregation. *b + +* flies were PCR-genotyped for the presence (row 1) or absence (row 4) of *Sd-RanGAP*.

In African populations, chromosomes bearing *Sd* but lacking *In(2R)Mal* do not drive (**Presgraves et al., 2009**; **Brand et al., 2015**). The functional role of *In(2R)Mal* in drive is, however, unclear. As expected, *In(2R)Mal* suppresses recombination: in crosses between a multiply marked chromosome *2*, *b c px*, and *SD-Mal* (**Figure 1—figure supplement 4**), we find that *In(2R)Mal* reduces the *b–c* genetic distance by 54.6% and the *c–px* genetic distance by 92.4%, compared with control crosses between *b c px* and Oregon-R (**Table 1**). Our crosses confirm that *In(2R)Mal* is indeed required for drive: if we generate recombinants along an *SD-Mal* chromosome, all recombinants with both *Sd* and *In(2R)Mal* show strong drive (**Table 2**, rows 1 and 2), whereas none of the recombinants that separate *Sd* and *In(2R)Mal* drive (**Table 2**, rows 3 and 4). We conclude that *SD-Mal* drive requires both *Sd* and *In(2R) Mal*, which implies that one or more essential enhancers, or co-drivers, is located within or distal to *In(2R)Mal*.

The temporal order of inversions (first *In(2R)Mal-d*, then *In(2R)Mal-p*) suggests two possible scenarios. *In(2R)Mal-d*, occurring first, may have captured the essential enhancer, with the subsequent *In(2R)Mal-p* serving to further reduce recombination between *Sd* and the enhancer. Alternatively, an essential enhancer may be located distal to *In(2R)Mal-d*, and the role of both *In(2R)Mal* inversions is to reduce recombination with *Sd*. To distinguish these possibilities, we measured drive in *b⁺ Sd c⁺ In(2R) Mal px* recombinants, which bear *Sd* and *In(2R)Mal* but have recombined between the distal breakpoint of *In(2R)Mal* (2R:18,774,475) and *px* (2R:22,494,297). All of these recombinants show strong drive (*n* = 71; **Table 2**, row 2). Assuming that recombination is uniformly distributed throughout the 3.72 Mb interval between the *In(2R)Mal-d* distal breakpoint and *px*, the probability of failing to separate an essential co-driver or distal enhancer among any of our 71 recombinants is <0.014. Furthermore, using molecular markers (see Materials and methods), we detected two recombinants within 100 kb of the distal breakpoint of *In(2R)Mal*, both with strong drive (*k* > 0.99; **Supplementary file 2**). We therefore infer that the co-driver resides inside or within 100 kb of the *In(2R)Mal* arrangement. More specifically, we speculate that the *In(2R)Mal-d* inversion both captured the co-driver and reduced recombination with *Sd*, whereas *In(2R)Mal-p* tightened linkage between centromere-proximal components of *SD-Mal* and *In(2R)Mal-d*.

Despite the recruitment of these inversions, recombination occurs readily between *Sd* and the proximal break of *In(2R)Mal* (**Table 2**; **Presgraves et al., 2009**; **Brand et al., 2015**). Nevertheless, we observe long-range linkage disequilibrium between *Sd* and *In(2R)Mal*. Among 204 haploid genomes from Zambia (**Lack et al., 2016**; see Materials and methods), we identified 198 wildtype haplotypes (*Sd⁺ In(2R)Mal⁺*), 3 *SD-Mal* haplotypes (*Sd In(2R)Mal*), and 3 recombinant haplotypes (three *Sd In(2R) Mal⁺*, zero *Sd⁺ In(2R)Mal*). While *Sd* and *In(2R)Mal* each have individually low sample frequencies

**Table 3.** Nucleotide diversity ($\pi$) on *SD-Mal* and *SD⁺* chromosomes.

| | | | $\pi$ ( ± st. dev.) | | | p-value | |
|---|---|---|---|---|---|---|---|
| | Chr. | Region | SD⁺ | SD-Mal | SD⁺ × f | SD-Mal vs. SD⁺ | SD-Mal vs. SD⁺*f |
| 1 | *2L* | Distal to *Sd-RanGAP* | 1.03E-02 (±3.01E-03) | 1.03E-02 (±3.09E-03) | 1.52E-04 (±4.43E-05) | 0.5727 | 0.00E + 00 |
| 2 | *2L* | Proximal to *Sd-RanGAP* | 4.44E-03 (±2.75E-03) | 9.39E-05 (±1.66E-04) | 6.52E-05 (±4.04E-05) | 5.84E-90 | 0.0027 |
| 3 | *2R* | *In(2R)Mal* | 8.94E-03 (±2.95E-03) | 7.97E-05 (±1.18E-04) | 1.31E-04 (±4.33E-05) | 0.00E + 00 | 1.42E-33 |
| 4 | *2L-2R* | *SD-Mal* supergene | 6.42E-03 (±4.03E-03) | 7.98E-05 (±1.32E-04) | 9.43E-05 (±5.92E-05) | 0.00E + 00 | 2.60E-06 |

Average nucleotide diversity ($\pi$) per site and empirical standard deviation estimated in 10-kb windows along chromosome 2, for *SD⁺*, *SD-Mal*, and *SD⁺* scaled by the estimated frequency of *SD-Mal* chromosomes (*SD⁺*× f, where *f* = 1.47%). Outside of the linked region (row 1), $\pi_{SD\text{-}Mal} \sim \pi_{SD}^+$. Inside of the linked region (rows 2–4), $\pi_{SD\text{-}Mal} < \pi_{SD}^+$; even after scaling $\pi_{SD}^+$ by the frequency of *SD-Mal* in the population, $\pi_{SD\text{-}Mal} < \pi_{SD}^+ \times f$. Due to non-independence of SNPs in non-recombining regions, we also estimated variance in $\pi$ based on **Charlesworth and Charlesworth, 2010**; which is 5.30E-05 for *In(2R)Mal* and 6.27E-05 for the entire *SD-Mal* supergene. p-values reported by paired *t*-test between 10-kb windows.

(0.0294 and 0.0147, respectively), they tend to co-occur on the same chromosome ($r^2$ = 0.493; Fisher's exact p = 1.4 × 10$^{-5}$). We calculated the expected decay of linkage disequilibrium between *Sd* and *In(2R)Mal* in the absence of any natural selection (*Hill and Robertson, 1968*), assuming a conservative sex-averaged recombination frequency corresponding to a map distance between *Sd* and *In(2R)Mal* of ~2.5 cM (FlyBase; FB2021_06; *Larkin et al., 2021*) and an effective population size of 10$^6$. Under these assumptions, the observed estimated coefficient of linkage disequilibrium, *D* = 0.0143, has an expected half-life of just ~28 generations (2.8 years) and, decays to negligible levels (i.e., expected *D* and $r^2$ both ~10$^{-3}$) in <100 generations (<10 years). We therefore conclude that the *SD-Mal* supergene haplotype is maintained by strong epistatic selection.

## Rapid increase in frequency of the *SD-Mal* supergene

We used population genomics to infer the evolutionary history and dynamics of *SD-Mal* chromosomes. We called SNPs in our Illumina reads from nine complete *SD-Mal* haplotypes from Zambia

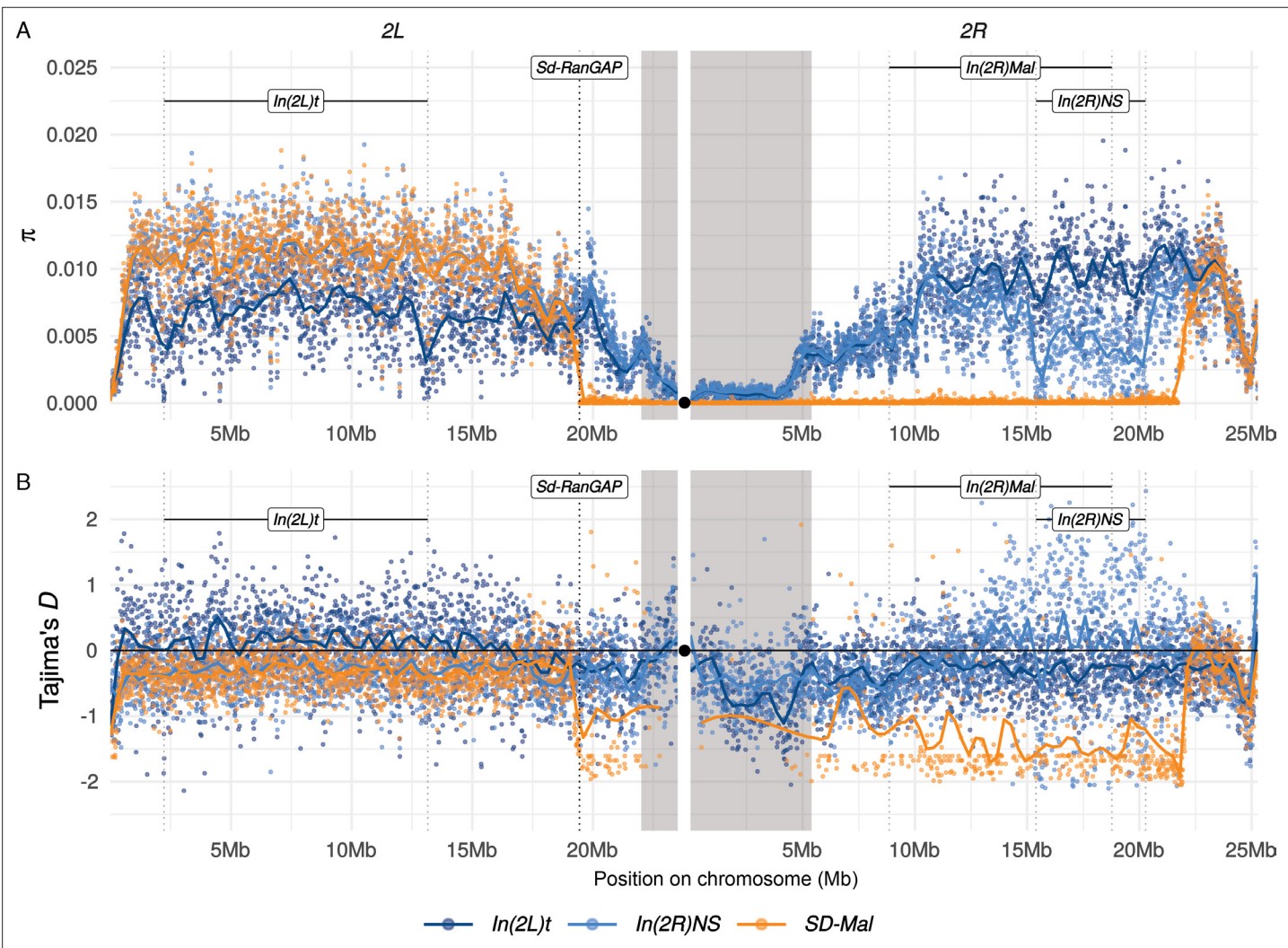

**Figure 2.** Diversity on *SD-Mal* chromosomes. (**A**) Average pairwise nucleotide diversity per site (*π*) and (**B**) Tajima's *D* estimates in non-overlapping 10-kb windows along chromosome 2 in Zambian *SD-Mal* chromosomes (*n* = 9, orange) and *SD$^+$* chromosomes from the same population, bearing the cosmopolitan inversions *In(2L)t* (*n* = 10, dark blue) and *In(2R)NS* (*n* = 10, light blue). Regions corresponding to pericentric heterochromatin are shaded in gray and the centromere location is marked with a black circle. *SD-Mal* chromosomes show a sharp decrease in nucleotide diversity and skewed frequency spectrum from the *Sd* locus (*Sd-RanGAP*, *2L*:19.4 Mb) to ~2.9 Mb beyond the distal breakpoint of *In(2R)Mal*.

The online version of this article includes the following figure supplement(s) for figure 2:

**Figure supplement 1.** Diversity at the *Sd-RanGAP* locus on *SD-Mal* chromosomes.

**Figure supplement 2.** Tajima's *D* estimates.

(see Materials and methods). For comparison, we also analyzed wildtype ($SD^+$) chromosomes from the same population in Zambia (*Lack et al., 2016*), including those with chromosome *2* inversions: 10 with the *In(2L)t* inversion and 10 with the *In(2R)NS* inversion (see Materials and methods). *Table 3* shows that nucleotide diversity ($\pi$) is significantly lower on *SD-Mal* haplotypes compared to uninverted $SD^+$ chromosome arms (*Table 3*; *Figure 2A*). The relative reduction in diversity on *SD-Mal* haplotypes is distributed heterogeneously: $\pi$ is sharply reduced for a large region that spans ~25.8 Mb, representing 53% of chromosome 2 and extending from *Sd-RanGAP* on *2L* (*2L*:19,441,959; *Figure 2—figure supplement 1*), across the centromere, and to ~2.9 Mb beyond the distal breakpoint of *In(2R)Mal* (*2R*:18,774,475; *Table 3*, rows 3, 5, and 6; *Figure 2A*). Thus, the region of reduced nucleotide diversity on *SD-Mal* chromosomes covers all the known essential loci for the drive phenotype: *Sd-RanGAP*, *Rsp^i^*, and *In(2R)Mal*.

The reduced nucleotide diversity among *SD-Mal* might be expected given its low frequency in natural populations (see below; *Presgraves et al., 2009*; *Brand et al., 2015*). *SD* persists at low frequencies in populations worldwide, presumably reflecting the balance between drive, negative selection, and genetic suppression and/or resistance (*Hartl, 1975*; *Charlesworth and Hartl, 1978*; *Larracuente and Presgraves, 2012*). If the *SD-Mal* supergene has been maintained at stable drive-selection-suppression equilibrium frequency for a long period of time, then its nucleotide diversity may reflect a mutation-drift equilibrium appropriate for its effective population size. Under this scenario, we expect diversity at the supergene to be similar to wildtype ($SD^+$) diversity scaled by the long-term equilibrium frequency of *SD*. We estimated *SD-Mal* frequency to be 1.47% by identifying the *Sd* duplication and *In(2R)Mal* breakpoints in 204 haploid genomes from Zambia (3/204, comparable to *Presgraves et al., 2009*; *Brand et al., 2015*; data from *Lack et al., 2016*; see Materials and methods). To approximate our expectation under mutation-drift equilibrium, we scaled average $\pi$ from the $SD^+$ sample by 1.47% in 10-kb windows across the region corresponding to the *SD-Mal* supergene, defined as the region from *Sd-RanGAP* to the distal breakpoint of *In(2R)Mal*. While nucleotide diversity outside of the *SD-Mal* supergene region is comparable to $SD^+$ (*Table 3*, row 1), diversity in the supergene region is significantly lower than expected even when scaled by its frequency (*Table 3*, row 4), suggesting that the low population frequency of *SD-Mal* cannot fully explain its reduced diversity. This observation suggests two possibilities: the *SD-Mal* supergene historically had an equilibrium frequency less than 1.47% in Zambia; or the *SD-Mal* supergene, having reduced recombination, has experienced hitchhiking effects due to background selection and/or a recent selective sweep.

To distinguish between these possibilities, we analyzed summaries of the site frequency spectrum. We find strongly negative Tajima's D mirroring the distribution of reduced diversity, indicating an excess of rare alleles (*Figure 2B*). Such a skew in the site frequency spectrum suggests a recent increase in frequency of the *SD-Mal* supergene in Zambia. Given the low recombination frequency between *SD-Mal* and $SD^+$ chromosomes, we treat them as two subpopulations and estimate their differentiation using Wright's fixation index, $F_{ST}$. The high differentiation of *SD-Mal* from $SD^+$ chromosomes from the same population similarly suggests a large shift in allele frequencies. $F_{ST}$ in the *SD-Mal* supergene region is unusually high for chromosomes from the same population (*Figure 3A*). Neither of the $SD^+$ chromosomes with cosmopolitan inversions show such high differentiation, and mean nucleotide differences ($d_{XY}$) between *SD-Mal* and $SD^+$ are comparable to the other inversions, implying that the differentiation of the *SD-Mal* supergene is recent. Our results—low diversity, strongly negative Tajima's D, high $F_{ST}$ and relatively low $d_{XY}$—are thus consistent with a rapid increase in frequency of the *SD-Mal* haplotype that reduced nucleotide diversity within *SD-Mal* and generated large differences in allele frequencies with $SD^+$ chromosomes (e.g., *Charlesworth, 1998*; *Cruickshank and Hahn, 2014*).

To estimate the timing of the recent expansion of the *SD-Mal* supergene, we used an approximate Bayesian computation (ABC) method with rejection sampling in neutral coalescent simulations. We do not know if *SD* chromosomes acquired *In(2R)Mal* in Zambia or if the inversions occurred de novo on an *SD* background. For our simulations, we assume that the acquisition of the second inversion (or the double inversion by crossover) was a unique event that enhanced drive strength and/or efficiency and that the onset of the selective sweep occurred following this event. Under this scenario, extant *SD-Mal* chromosomes have a single origin. We therefore simulated this history in a coalescent framework as an absolute bottleneck to a single chromosome. We performed simulations considering a sample size of $n = 9$ and assumed no recombination in the ~9.92 Mb region of *In(2R)Mal*. We simulated with values of S drawn from a uniform distribution ±5% of the observed number of segregating sites in

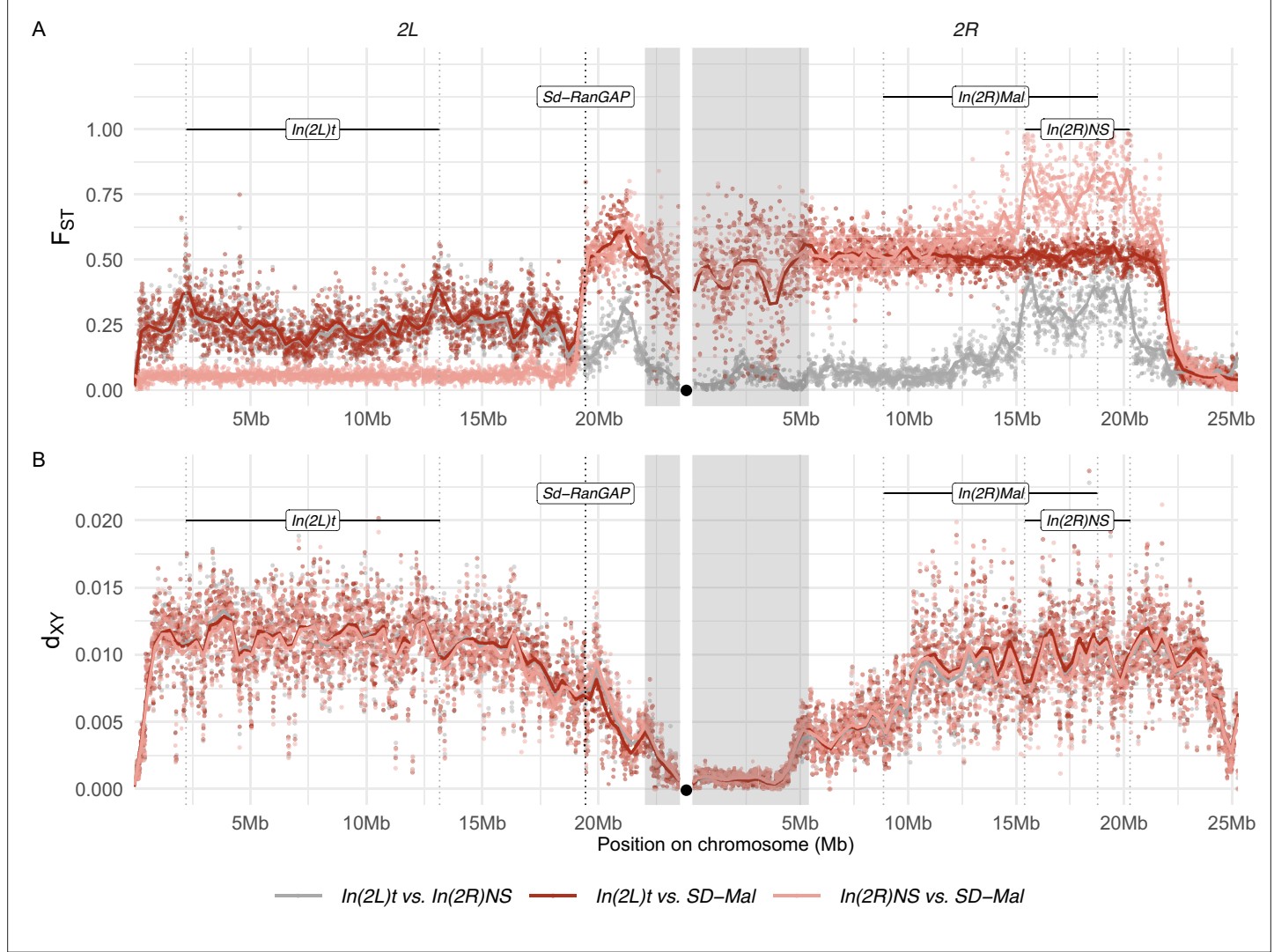

**Figure 3.** Differentiation between *SD-Mal* and wildtype chromosomes. (**A**) Pairwise $F_{ST}$ and (**B**) $d_{XY}$ per base pair in non-overlapping 10-kb windows along chromosome *2*, between Zambian *SD-Mal* haplotypes (*n* = 9) and wildtype chromosomes from the same population, bearing the cosmopolitan inversions *In(2L)t* (*n* = 10) and *In(2R)NS* (*n* = 10). Regions corresponding to pericentric heterochromatin are shaded in gray and the centromere location is marked with a black circle.

non-coding regions of *In(2R)Mal*. We considered a prior uniform distribution of the time of the expansion (*t*) ranging from 0 to $4N_e$ generations (0–185,836 years ago), assuming that *D. melanogaster* $N_e$ in Zambia 3,160,475 (**Kapopoulou et al., 2018**), a *In(2R)Mal* frequency of 1.47%, and 10 generations per year (**Li and Stephan, 2006**; **Thornton and Andolfatto, 2006**; **Laurent et al., 2011**; **Kapopoulou et al., 2018**). Using the ABC with rejection sampling conditional on our observed estimates of *π* and Tajima's *D* for *In(2R)Mal* ($\pi_{In(2R)Mal}$ = 584.60, *D* = –1.33; note that $\pi_{In(2R)Mal}$ is an overall, unscaled estimate of nucleotide diversity for the whole *In(2R)Mal* region and that only non-coding regions were considered), we infer that the *SD-Mal* expansion began ~0.0884 (95% CIs 0.0837–0.1067) × $4N_e$ generations ago or, equivalently, ~1644 years ago (1.11% rejection sampling acceptance rate; **Figure 4**). To account for possible effects of gene conversion between *SD* and $SD^+$ chromosomes (see below), we discarded SNPs shared with $SD^+$ chromosomes (see below), and recalculated *π* and Tajima's *D* using only private SNPs ($\pi_{In(2R)Mal}$ = 427.72, *D* = –1.45). Based on these parameters, the estimated *SD-Mal* expansion occurred ~0.0679 (95% CIs 0.0647–0.0868) $4N_e$ generations ago, ~1261 years (1.02% rejection sampling acceptance rate; **Figure 4**). To calculate the posterior probability of the model, we performed 100,000 simulations under three models: a model assuming a stable frequency of *SD-Mal*; a model assuming an exponential growth of *SD-Mal*, based on parameters estimated for

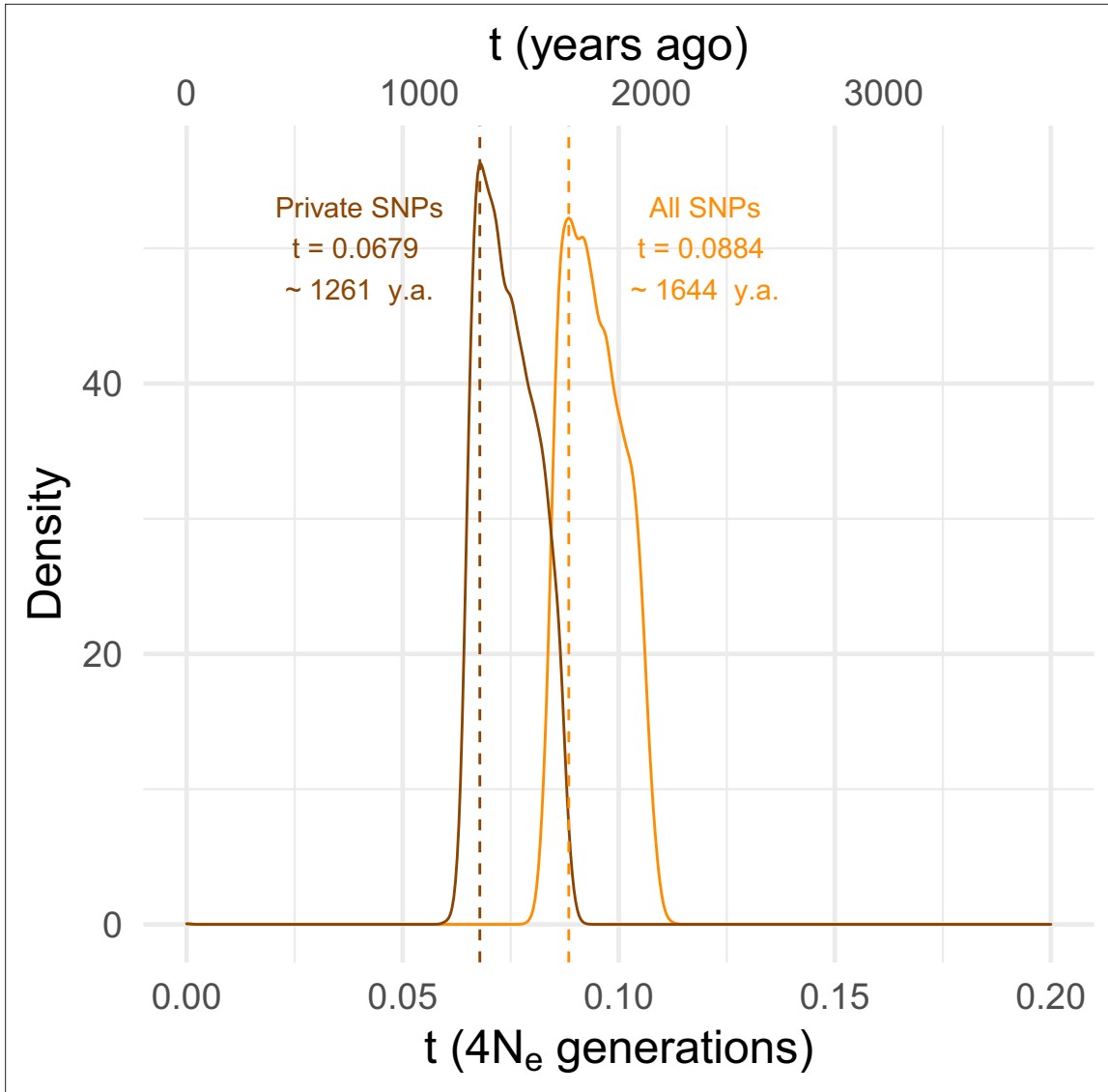

**Figure 4.** Estimating the time since the *SD-Mal* selective sweep. Approximate Bayesian computation (ABC) estimates based on 10,000 posterior samples place the onset of the selective sweep between 0.0884 (95% CI 0.0837–0.1067) and 0.0679 (0.0647–0.0868) × $4N_e$ generations, that is, ~1261– 1644 years ago, considering recent estimates of $N_e$ in Zambia from ***Kapopoulou et al., 2018***, frequency of *SD-Mal* in Zambia 1.47% and 10 generations per year. Estimates were done considering only *In(2R)Mal*, where crossing over is rare and only occurs between *SD-Mal* chromosomes, using all SNPs and excluding shared SNPs in order to account for gene conversion from *SD*⁺ chromosomes.

The online version of this article includes the following figure supplement(s) for figure 4:

**Figure supplement 1.** Neutral coalescent simulations under three demographic models.

Zambia (***Kapopoulou et al., 2018***); and a selective sweep model (assuming $t_{all}$ = 0.0884 and $t_{shared\_excl}$ = 0.0679) (***Figure 4—figure supplement 1***). The simulated data are inconsistent with a long-term stable frequency of *SD-Mal* (all SNPs, $p_\pi$ = 0.0522, $p_D$ = 0.096; private, $p_\pi$ = 0.0266, $p_D$ = 0.0668) or long-term exponential growth (all SNPs, $p_\pi$ = 0.0465, $p_D$ = 0.0907; private, $p_\pi$ = 0.0215, $p_D$ = 0.0605). Instead, our simulations suggest that a recent selective sweep is more consistent with the data (all SNPs, $p_\pi$ = 0.3554, $p_D$ = 0.5952; private, $p_\pi$ = 0.3480, $p_D$ = 0.6142). Taken together, evidence from nucleotide diversity, the site frequency spectrum, population differentiation, and coalescent simulations suggests a rapid non-neutral increase in frequency of the *SD-Mal* supergene that began <2000 years ago.

The sweep signal on the *SD-Mal* haplotypes begins immediately distal to *Sd-RanGAP* on *2L* and extends ~3 Mb beyond the distal boundary of *In(2R)Mal* on *2R*. To understand why the sweep extends so far beyond the *In(2R)Mal-d* distal breakpoint, we consider three, not mutually exclusive,

possibilities. First, chromosomal inversions can suppress recombination ~1–3 Mb beyond their break-points (in both multiply inverted balancer chromosomes, [*Miller et al., 2016*; *Crown et al., 2018*; *Miller et al., 2018*] and natural inversions [*Stevison et al., 2011*; *Fuller et al., 2017*]), extending the size of the sweep signal. To determine the extent of recombination suppression caused by *In(2R)Mal*, we estimated recombination rates in the region distal to the inversion. The expected genetic distance between the distal breakpoint of *In(2R)Mal* (*2R*:18.77 Mb) and *px* (*2R*:22.49 Mb) is ~13.87 cM (*Fiston-Lavier et al., 2010*). Measuring recombination between *SD-Mal* and standard arrangement chromosomes for the same (collinear) interval, we estimate a genetic distance of ~1.76 (*Table 1*), an 87.3% reduction. *In(2R)Mal* strongly reduces recombination beyond its distal boundary. Second, although we have inferred that the essential enhancer(s) reside(s) within the *In(2R)Mal* inversion (see above), we have not excluded the possibility of weak enhancers distal to the inversion which might contribute to the sweep signal. We find that *SD-Mal* chromosomes with *In(2R)Mal*-distal material recombined away (*b⁺ Sd c⁺ In(2R)Mal px*) have modestly but significantly lower drive strength ($k$ = 0.96 vs. 0.98; *Table 2*, lines 1–2), suggestive of one or more weak distal enhancers. Third, there may be mutations distal to *In(2R)Mal* that contribute to the fitness of *SD-Mal* haplotypes but without increasing the strength of drive, for example, compensatory mutations that ameliorate the effects of *SD-Mal*-linked deleterious mutations.

Most supergenes show long-range LD, reduced nucleotide diversity, and differentiation when compared with their wildtype counterparts. While some meiotic drive supergenes show evidence of recurrent selective sweeps (*Dyer et al., 2007*) or a signature of epistatic selection without strong selective sweeps (*Fuller et al., 2020*), others show no signatures of recent or ongoing positive selection (*Kelemen and Vicoso, 2018*). The relatively recent origin (~38.5 kya; *Brand et al., 2015*) of *SD* might explain the constant turnover, as there may not have been enough time to reach a stable equilibrium compared to older drive systems like the *t-haplotype*, whose first inversion arose 3 mya (*Hammer and Silver, 1993*).

## Recombination on *SD-Mal* supergenes

While nearly all *SD-Mal* haplotypes are individually homozygous lethal and do not recombine with wildtype chromosomes in and around *In(2R)Mal*, ~90% of pairwise combinations of different *SD-Mal* chromosomes ($SD_i/SD_j$) are viable and fertile in complementation tests (*Presgraves et al., 2009*; *Brand et al., 2015*). Therefore, recombination via crossing over may occur between *SD-Mal* chromosomes in $SD_i/SD_j$ heterozygous females. To determine if *SD-Mal* chromosomes recombine, we estimated mean pairwise linkage disequilibrium ($r^2$) between SNPs located within the *In(2R)Mal* arrangement. We found that mean $r^2$ between pairs of SNPs declines as a function of the physical distance separating them (*Figure 5A*), a hallmark of recombination via crossing over (*Hill and Robertson, 1968*; *Miyashita and Langley, 1988*; *Schaeffer and Miller, 1993*; *Awadalla et al., 1999*; *Conway et al., 1999*). Pairwise LD is higher and extends further in *In(2R)Mal* than in the equivalent region of *SD⁺* chromosomes or in any of the other two cosmopolitan inversions, *In(2L)t* and *In(2R)NS* (*Figure 5A*). This pattern is not surprising: the low frequency of *SD-Mal* makes $SD_i/SD_j$ genotypes, and hence the opportunity for recombination, rare. (The smaller sample size of *SD* ($n$ = 9) vs. *SD⁺* ($n$ = 10) may also contribute weakly to its higher estimated LD.) To further characterize the history of recombination between *SD-Mal* haplotypes, we used 338 non-singleton, biallelic SNPs in *In(2R)Mal* to trace historical crossover events. From these SNPs, we estimate that Rm (*Hudson and Kaplan, 1985*), the minimum number of recombination events, in this sample of *SD-Mal* haplotypes is 15 (*Figure 5C*). Thus, assuming that these *SD-Mal* haplotypes are ~16,436 generations old (*Figure 4*), we estimate that recombination events between *SD-Mal* haplotypes occur a minimum of once every ~1096 generations. We can thus confirm that crossover events are relatively rare, likely due to the low population frequency of *SD-Mal* and the possibly reduced fitness of $SD_i/SD_j$ genotypes.

While crossing over is suppressed in *SD-Mal/SD⁺* heterozygotes, gene conversion and/or double crossover events may still occur, accounting for the shared SNPs between *SD-Mal* and *SD⁺* chromosomes within *In(2R)Mal*. As both events exchange tracts of sequence, we expect shared SNPs to occur in runs of sites at higher densities than private SNPs, which should be distributed randomly. Accordingly, in *In(2R)Mal*, SNP density is five times higher for runs of shared SNPs (0.63 SNPs/kb) than for runs of *SD*-private SNPs (0.12 SNPs/kb), as expected if *SD⁺* chromosomes, which have higher SNP densities, were donors of conversion tract sequences. Although we cannot exclude the contribution of

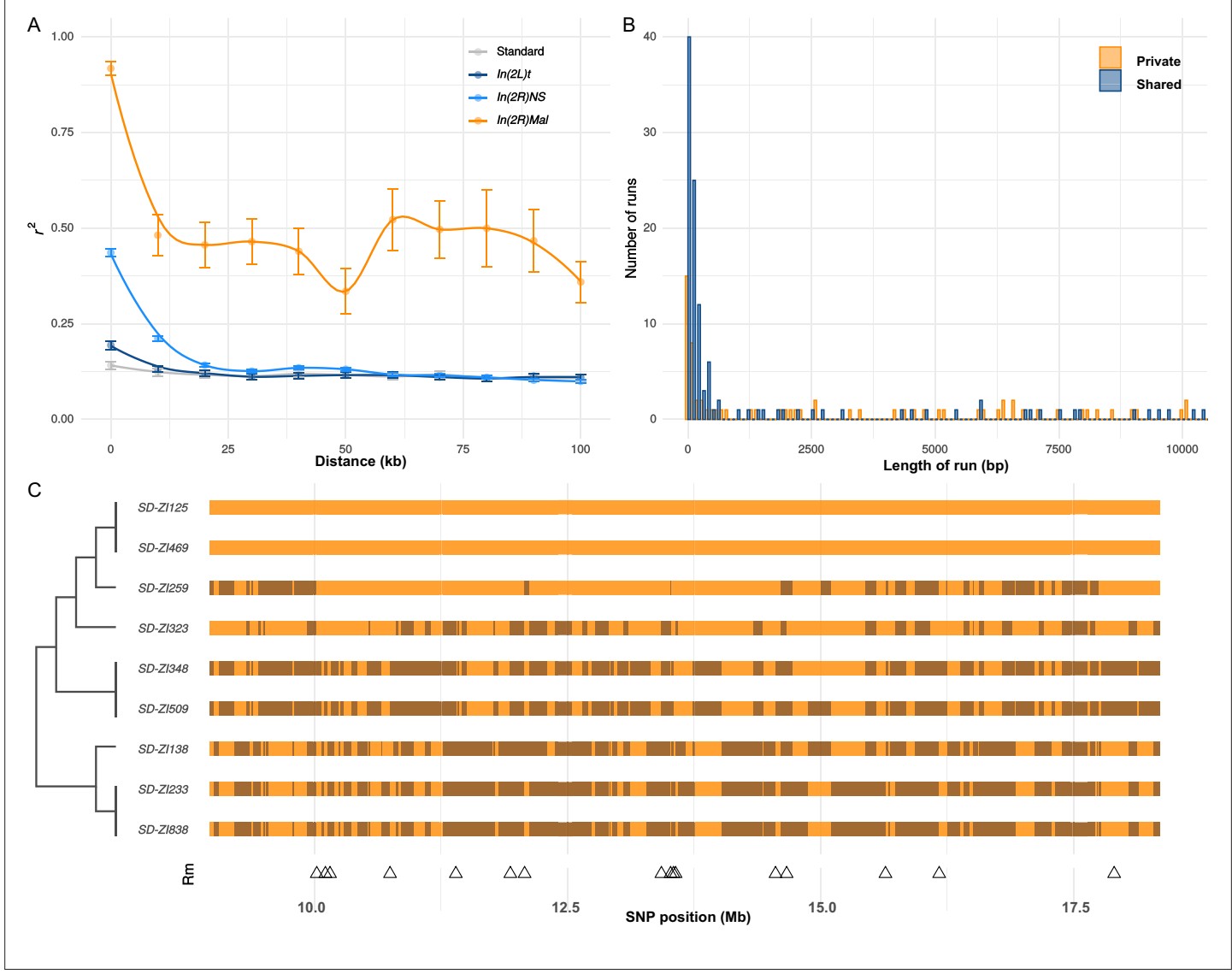

**Figure 5.** Recombination on *SD-Mal* haplotypes. (**A**) Linkage disequilibrium (*r²*) as a function of distance in 10-kb windows, measured in *In(2R)Mal* (*n* = 9), *In(2L)t* (*n* = 10), *In(2R)NS* (*n* = 10), and the corresponding region of *In(2R)Mal* in a standard, uninverted *2R* chromosome (*n* = 10). (**B**) Histogram of length of runs of SNPs in *In(2R)Mal* shows that a high proportion of shared SNPs concentrate in runs shorter than 1 kb. (**C**) Chromosomal configuration of the 338 non-singleton SNPs in nine different *SD-Mal* lines. Color coded for two states (same in light orange or different in dark orange) using *SD-ZI125* as reference. Locations of minimal number of recombination events are labeled as triangles at the bottom. Maximum likelihood tree is displayed on the left.

double crossovers, we note that 62.2% (89 out of 143) of the shared SNP runs are <1 kb, 80.4% (115 out of 143) are <10 kb (*Figure 5B*), and the longest run is ~50.2 kb. These sizes are more consistent with current estimates of gene conversion tract lengths in *D. melanogaster* than with double crossovers (*Comeron et al., 2012*; *Miller et al., 2016*). Surprisingly, these inferred gene conversion events are unevenly distributed across *In(2R)Mal*, being more frequent in the *In(2R)Mal-p* than in *In(2R)Mal-d* (*Supplementary file 3*). Our discovery that *SD-Mal* haplotypes can recombine with each other distinguishes the *SD-Mal* supergene from supergenes that are completely genetically isolated (*Wang et al., 2013*; *Charlesworth, 2016*; *Tuttle et al., 2016*). The lack of crossing over with *SD⁺* chromosomes, however, means that *SD-Mal* haplotypes evolve as a semi-isolated subpopulation, with a nearly 100-fold smaller $N_e$ and limited gene flow from *SD⁺* via gene conversion events. The reduced recombination, low $N_e$, and history of epistatic selection may nevertheless lead to a higher genetic load on

**Table 4.** Synonymous and non-synonymous SNPs.

|  | Genotype | N | S | N/S | Fold change | p-value |
|---|---|---|---|---|---|---|
| All SNPs | *SD-Mal* | 79 | 114 | 0.69 | | |
| | *SD*[+] | 10,470 | 34,301 | 0.31 | 2.27 | <0.0001 |
| Private SNPs | *SD-Mal* | 61 | 55 | 1.11 | | |
| | *SD*[+] | 6782 | 18,938 | 0.36 | 3.10 | <0.0001 |
| Shared SNPs | *SD-Mal* | 18 | 59 | 0.31 | | |
| | *SD*[+] | 3688 | 15,363 | 0.24 | 1.27 | 0.3722 |

Counts of non-synonymous (*N*) and synonymous (*S*) SNPs in the *In(2R)Mal* region of *SD-Mal* chromosomes, and the equivalent region of uninverted, *SD*[+] chromosomes. *N/S* ratio per genotype, fold-change of *N/S* ratios between *SD-Mal* and *SD*[+]. p-values reported by Pearson's $\chi^2$ test of independence.

*SD-Mal* than *SD*[+] chromosomes. We therefore examined the accumulation of deleterious mutations, including non-synonymous mutations and TEs, on the *SD-Mal* supergene.

## Consequences of reduced recombination, small effective size, and epistatic selection

We first studied the effects of a reduced efficacy of selection on SNPs in *In(2R)Mal*. As many or most non-synonymous polymorphisms are slightly deleterious (*Ohta, 1976*; *Fay et al., 2001*; *Eyre-Walker et al., 2002*; *Keightley and Eyre-Walker, 2007*; *Eyre-Walker and Keightley, 2009*), relatively elevated ratios of non-synonymous to synonymous polymorphisms (*N/S* ratio) can indicate a reduced efficacy of negative selection. For the SNPs in *In(2R)Mal*, the overall *N/S* ratio is 2.3-fold higher than that for the same region of *SD*[+] chromosomes (*Table 4*). Notably, the *N/S* ratio for private SNPs is 3.1-fold higher (*Table 4*), whereas the *N/S* ratios for shared SNPs do not significantly differ from *SD*[+] chromosomes (*Table 4*, *Figure 6—figure supplement 1*). These findings suggest that gene conversion from *SD*[+] ameliorates the accumulation of potentially deleterious non-synonymous mutations on *SD-Mal* chromosomes (see also *Kelemen and Vicoso, 2018*; *Pieper and Dyer, 2016*).

Gene conversion may not, however, rescue *SD-Mal* from deleterious TEs insertions, as average TE length exceeds the average gene conversion tract length (*Kaminker et al., 2002*). TEs accumulate in regions of reduced recombination, such as centromeres (*Charlesworth et al., 1994*) and inversions, especially those at low frequency (*Eanes et al., 2009*; *Sniegowski and Charlesworth, 1994*). Indeed, TE densities for the whole euchromatic region of chromosome *2R* are significantly higher for *SD-Mal* compared to *SD*[+] chromosomes (*Figure 6A*). This increased TE density on *SD-Mal* is driven by the low recombination regions of the haplotype: *In(2R)Mal* has significantly higher TE density than *SD*[+] whereas the distal region of *2R* outside of the sweep region does not (*Figure 6B*). The most over-represented families in *In(2R)Mal* relative to standard chromosomes are *M4DM*, *MDG*1, *ROO_I*, and *LINE* elements (*Figure 6—figure supplement 2*)—TEs that are currently or recently active (*Kaminker et al., 2002*; *Kofler et al., 2015*; *Diaz-González and Dominguez, 2020*)—consistent with the recent origin of the *SD-Mal* haplotype. Thus, the differences in shared vs. private SNPs suggests that gene conversion from *SD*[+] chromosomes may slow the accumulation of deleterious point mutations but not the accumulation of TEs. Despite occasional recombination, the small $N_e$ of *SD-Mal* haplotypes has incurred a higher genetic load.

## Conclusions

Supergenes are balanced, multigenic polymorphisms. Under the classic model of supergene evolution, epistatic selection among component loci favors the recruitment of recombination modifiers that reinforce the linkage of beneficial allelic combinations. The advantages of reduced recombination among strongly selected loci can however compromise the efficacy of selection at linked sites. Supergenes thus provide opportunities to study the interaction of recombination and natural selection. We have studied a population of *selfish* supergenes, the *SD-Mal* haplotypes of Zambia, to investigate the interplay of recombination, selection, and meiotic drive. Our findings demonstrate, first, that the *SD-Mal* supergene extends across ~25.8 Mb of *D. melanogaster* chromosome *2*, a

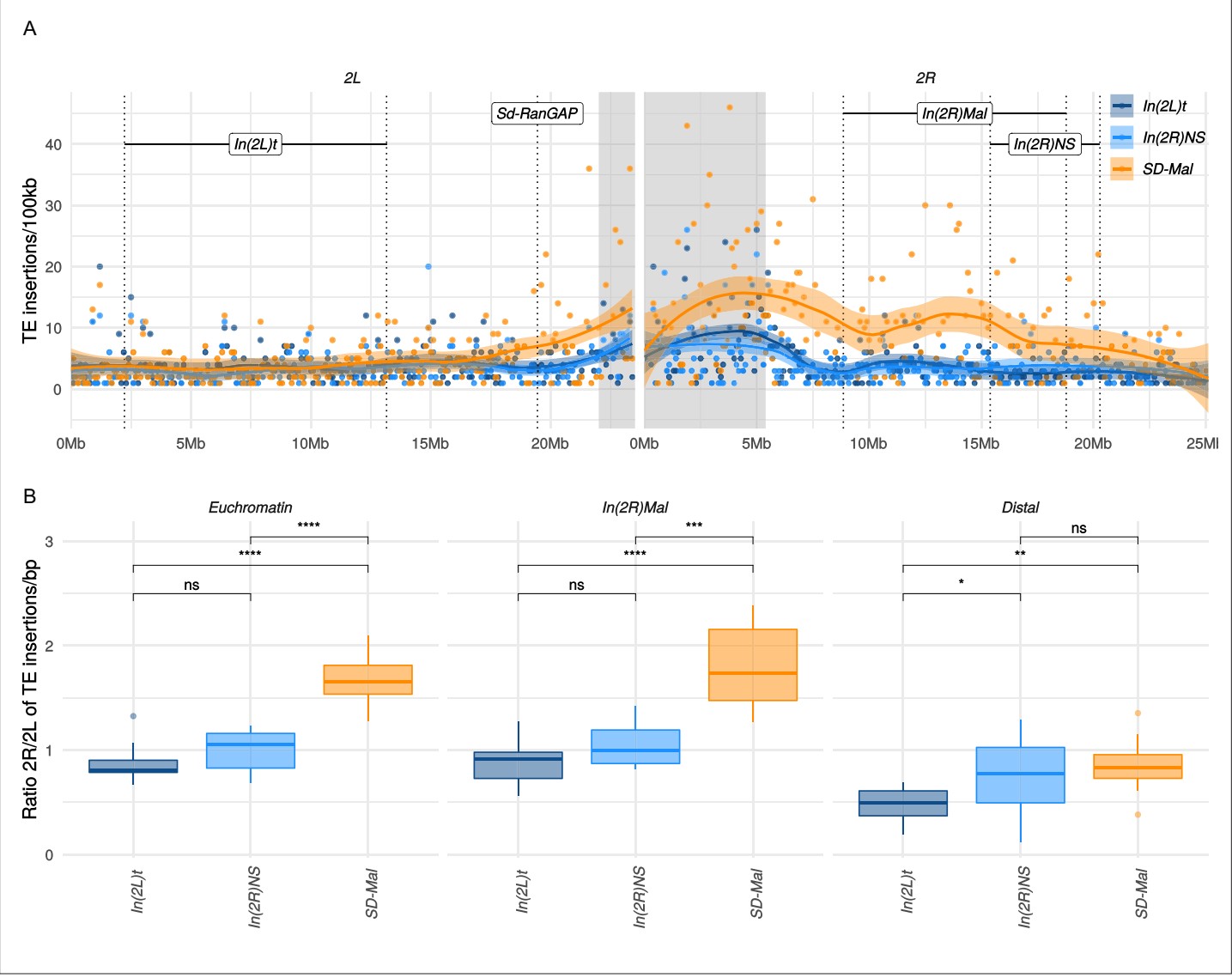

**Figure 6.** Transposable elements (TEs) on *SD-Mal* haplotypes. (**A**) Number of *TE* insertions per 100-kb windows along chromosome 2 in Zambian *SD* chromosomes (*n* = 9, orange) and wildtype chromosomes from the same population, bearing the cosmopolitan inversions *In(2L)t* (*n* = 10, dark blue) and *In(2R)NS* (*n* = 10, light blue). (**B**) Ratio of the number of insertions in the euchromatin of *2R* to *2L* per library. The relative enrichment in TEs in *2R* of *SD-Mal* haplotypes is mostly due to an increase of TE insertions in non-recombining regions of the chromosome. Asterisks denote significance, p-values estimated by a Kruskal-Wallis test (threshold for significance p = 0.05).

The online version of this article includes the following figure supplement(s) for figure 6:

**Figure supplement 1.** Frequency spectra of synonymous and non-synonymous SNPs.

**Figure supplement 2.** Number of insertions per transposable element (TE) family.

**Figure supplement 3.** Abundance of transposable elements (TEs) in downsampled libraries.

region that comprises the driving *Sd-RanGAP*, a drive-insensitive deletion at the major *Rsp* locus, and the *In(2R)Mal* double inversion. Second, using genetic manipulation, we show that *SD-Mal* requires *Sd-RanGAP* and an essential co-driver that localizes almost certainly within the *In(2R)Mal* rearrangement, and probably within the distal inversion. These data provide experimental evidence for epistasis between *Sd-RanGAP* and *In(2R)Mal*: neither allele can drive without the other. Third, we provide population genomics evidence that epistatic selection on loci spanning the *SD-Mal* super-gene region drove a very recent, chromosome-scale selective sweep. These patterns are consistent with recurrent episodes of replacement of one *SD* haplotype by others (***Presgraves et al., 2009***;

*Brand et al., 2015*). Fourth, despite rare crossovers among complementing *SD-Mal* haplotypes and gene conversion from wildtype chromosomes, the relative genetic isolation and low frequency of *SD-Mal* results in the accumulation of deleterious mutations including, especially, TE insertions. From these findings, we conclude that the *SD-Mal* supergene population is of small effective size, semi-isolate from the greater population of wildtype chromosomes, and subject to bouts of very strong selection.

Non-recombining supergenes that exist exclusively in heterozygous state tend to degenerate, as in the case of Y chromosomes (reviewed in *Charlesworth and Charlesworth, 2000*) and some autosomal supergenes which, for different reasons, lack any opportunity for recombination (*Uyenoyama, 2005*; *Wang et al., 2013*; *Tuttle et al., 2016*; *Branco et al., 2018*; *Stolle et al., 2019*; *Brelsford et al., 2020*). But not all supergenes are necessarily expected to degenerate. In *SD-Mal*, for instance, complementing *SD-Mal* haplotypes can recombine via crossing over, if rarely, and gene flow from wildtype *SD$^+$* to *SD-Mal* chromosomes can occur via gene conversion. In the mouse *t*-haplotype, there is similar evidence for occasional recombination between complementing *t*-haplotypes (*Dod et al., 2003*) and with standard chromosomes, probably via gene conversion (*Herrmann et al., 1987*; *Erhart et al., 2002*; *Wallace and Erhart, 2008*; *Kelemen and Vicoso, 2018*). Despite the many parallels characterizing supergenes, their ultimate evolutionary fates depend on the particulars of the system.

## Materials and methods
### Fly lines, library construction, and sequencing

We sequenced haploid embryos using the scheme in *Langley et al., 2011*, which takes advantage of a mutation, *ms(3)K81* (*Fuyama, 1984*), which causes the loss of the paternal genome early in embryonic development. We crossed *SD-Mal/CyO* stocks generated in *Brand et al., 2015* to homozygous *ms(3)K81* males and allowed them to lay eggs overnight. We inspected individual embryos under a dissecting scope for evidence of development and then isolated them for whole genome amplification using the REPLI-g Midi kit from Qiagen (catalog number 150043). For each WGA DNA sample, we tested for the presence of *Sd-RanGAP* using PCR (primers from *Presgraves et al., 2009*). We prepared sequencing libraries for Illumina sequencing with TruSeq PCR free 350 bp. We assessed library quality using a BioAnalyzer and sequenced with HiSeq2500 2 × 150 bp reads (TruSeq) or 2 × 125 bp reads (Nextera). To trim reads, we used Trimgalore v0.3.7 and the parameters: *q 28 --length 20 --paired -a GATCGGAAGAGCACACGTCTGAACTCCAGTCAC -a2 GATCGGAAGAGCGTCGTGTA GGGAAAGAGTGT --phred33 --fastqc --retain_unpaired -r1 21 r2 21 --dont_gzip --length 20*. Trimmed reads are available in SRA (Bioproject PRJNA649752, SRA accession numbers in *Supplementary file 1*, Sheet 1).

We sequenced a total of 10 *SD-Mal* genomes. One of these genomes (*SD-ZI157*) was *Sd-In(2R) Mal$^+$*, non-driving, and therefore excluded from further analysis. Out of the remaining nine driving *SD-Mal* chromosomes, one of them (*SD-ZI138*) had lower depth than the other eight (*Supplementary file 1*, Sheet 2) in the main chromosome arms but unusually high depth in the mitochondrial genome. We ran additional analyses dropping *SD-ZI138* and show that including this sample does not affect our main conclusions (*Supplementary file 5*; Sheet 2).

For the Nanopore library, we extracted High-Molecular-Weight DNA from ~200 frozen female *SD-ZI125/SD-ZI125* virgins. We extracted DNA using a standard phenol-chloroform method and spooled DNA using capillary tubes. We constructed a library with ~1 µg DNA using RAD004 kit and the ultra-long read sequencing protocol (*Quick, 2018*). We sequenced the library using R9.4 flow cells and called bases with the ONT Albacore Sequencing Pipeline Software version v2.2.10.

### Estimating *Rsp* copy number

We mapped Zambian *SD* reads to an assembly containing *2R* pericentric heterochromatin (*Chang and Larracuente, 2019*), including the *Rsp* locus detailed in *Khost et al., 2017*, with bowtie2 v2.3.5 (*Langmead and Salzberg, 2012*). We estimated mean per-window and per-*Rsp* repeat depth using mosdepth v0.2.9 (*Pedersen and Quinlan, 2018*). Coordinates for *Rsp* repeats were based on annotations in *Khost et al., 2017*.

## *In(2R)Mal* breakpoints

To assemble *SD-ZI125*, we filtered Nanopore reads using Porechop (v0.2.3) and Filtlong (--min_length 500) to remove adapters and short reads (https://github.com/rrwick/Porechop [*Wick et al., 2017*; *Wick, 2017a*] and https://github.com/rrwick/Filtlong [*Wick, 2017b*]). We were left with a total of 1,766,164,534 bases in 327,248 filtered reads. We performed de novo assemblies with the Nanopore reads using Flye v2.3.7 (*Kolmogorov et al., 2019*) with parameters '-t 24 g 160 m --nano-raw' and wtdbg v2.2 (*Ruan and Li, 2020*) with parameters '-p 19 -AS 1 s 0.05 L 0 -e 1'. We independently polished these two assemblies 10 times with Pilon v1.22 (*Walker et al., 2014*) using paired-end Illumina reads. Lastly, we reconciled these two polished assemblies using quickmerge v0.3 (*Chakraborty et al., 2016*) using the flye assembly as the reference with the command 'python merge_wrapper.py wtdbg assembly flye assembly'. We aligned the contig containing the euchromatin on *SD-ZI125* to chromosome *2R* of the *D. melanogaster* (BDGP6) genome using Mauve (*Darling et al., 2010*). We defined the breakpoints according to the block rearrangement shown in *Figure 1*. To validate these breakpoints, we designed primers anchored at both sides of the most external breakpoints of *In(2R) Mal* (*Supplementary file 4*) for PCR.

## Measuring genetic distances along *SD-Mal* and strength of distortion in the recombinants

To estimate recombination frequencies and obtain *SD-Mal* recombinant genotypes, we used a *D. melanogaster* stock *al[1] dpy[ov1] b[1] pr[1] c[1] px[1] sp[1]*, from Bloomington *Drosophila* Stock Center (RRID:BDSC_156), which has three visible, recessive markers on chromosome 2 that are informative about recombinants involving *SD: black* (*b*, 2L: 13.82), *curved* (*c*, 2R:15.9), and *plexus* (*px*, 2R:22.49). As our *SD* flies are white-eyed, we did not score *purple* (*pr,* an eye color phenotype). All crosses were transferred to fresh vials after 5 days, and then adults were removed from the second vial after 5 days. Progeny emerging from the crosses were scored for up to 20 days following the cross.

To generate *SD-Mal* recombinant chromosomes, we crossed 8–10 *b c px/b c px* virgin females to 3–5 *SD-ZI125* males, recovered *SD-ZI125/b c px* virgins, then backcrossed 8–10 of them to 3–5 *b c px* homozygous males (*Figure 1—figure supplement 4*). To estimate genetic distance between the visible markers, we scored the number of recombinants in 11 crosses (*n* = 1820). To compare genetic distance in *SD-Mal* to wildtype chromosomes, we estimated the number of recombinants from 15 crosses between *OregonR/b c px* females to *b c px/b c px* males (*n* = 1716).

We recovered three types of recombinant chromosomes from *SD-ZI125/b c px x b c px/b c px* crosses: $b\ Sd^+\ c^+\ In(2R)Mal\ px^+$; $b^+\ Sd\ c\ In(2R)Mal^+\ px$ and $b^+\ Sd\ c^+\ In(2R)Mal\ px$ (*Figure 1—figure supplement 4*). We crossed 3–5 virgin *b c px/b c px* females to individual recombinant males of each type, and scored the proportion of progeny carrying the recombinant chromosome ($k = n_{recombinant}/n_{total}$). To distinguish distortion from viability effects, we also measured transmission of recombinant chromosomes through females, as drive is male-specific. We used these crosses to measure relative viability ($w = n_{recombinant}/n_{bcpx}$). We then used $w$ to calculate a viability-corrected strength of distortion in males ($k^* = n_{recombinant}/(wn_{bcpx} + n_{recombinant})$) (*Powers and Ganetzky, 1991*).

To check for recombination close to the inversion breakpoints, we amplified two regions between the distal breakpoint of *In(2R)Mal* (*2R*:18:77) and *px* (*2R*:22.49), corresponding to the genes *sano* (*2R*:18.87; F primer: GGACCATTTCTAGGGCATCA, R primer: AATGAAACGTCCCCTCTTTG) and *CG15666* (*2R*:21.34; F primer: GGCCTATTGCGAGAGAACTG, R primer: TGCTTCCTTGATCTCG TCCT). Primers were designed so that they yield amplicons of different length in *SD-Mal* and *SD⁺*.

## Estimate of the frequency of *SD-Mal* in the DPGP3 dataset

To estimate the frequency of *In(2R)Mal* in a random sample of Zambian chromosomes, we mapped the 204 Illumina paired-end libraries from the DPGP3 dataset (*Lack et al., 2016*) to the *D. melanogaster* (BDGP6) genome, using bwa-mem (v0.7.9a), and we visually looked for an accumulation of discordant read pairs surrounding the estimated breakpoints of *In(2R)Mal*. To test the reliability of this method, we also applied it to detect cosmopolitan inversions *In(2L)t* and *In(2R)NS* and compared our inversion calls with the most recent inversion calls for the DPGP3 dataset (http://johnpool.net/Updated_Inversions.xls, last accessed 07/13/2020), getting a 98% and 99% of concordance for *In(2L)t* and *In(2R)NS*, respectively. To determine the frequency of the *Sd-RanGAP* duplication in the DPGP3 dataset, we

applied a similar method around the breakpoints of the *Sd-RanGAP* duplication (see ***Supplementary file 5***).

## SNP calling and annotation

For SNP calling, we mapped the Illumina reads from our *SD-Mal* libraries and the 20 *SD+* libraries from the *DPGP3* dataset to *D. melanogaster* (BDGP6) genome (ftp://ftp.ensembl.org/pub/release-88/fasta/drosophila_melanogaster/dna/; last accessed o6/25/20) using BWA mem (v0.7.9a). We removed duplicated reads with Picard (2.0.1) and applied the GATK (3.5) 'best practices' pipeline for SNP calling. We did local realignment and base score recalibration using SNPs data from DPGP1 ensembl release 88 (ftp://ftp.ensembl.org/pub/release-88/variation/vcf/drosophila_melanogaster/; last accessed 06/25/20). To call SNPs and indels, we used HaplotypeCaller and performed joint genotyping for each of the five genotypes using GenotypeGVCFs. SNPs filtered with following parameters: 'QD <2.0 || FS >60.0 || MQ <40.0 || MQRankSum <–12.5 || ReadPosRankSum <–8.0'. We annotated SNPs as synonymous or non-synonymous using SNPeff (4.3, ***Cingolani et al., 2012***) with the integrated *D. melanogaster* database (dmel_r6.12) database and parsed these annotations with SNPsift (***Cingolani et al., 2012***). To classify the SNPs as 'shared' between *SD-Mal*, *SD+In(2L)t* and *SD+In(2R)NS*, or 'private' to each one of them, we used BCFtools intersect (1.6; ***Danecek et al., 2021***).

## Population genomics analysis

We wrote a Perl script to estimate $S$, $\pi$, Tajima's $D$, $F_{ST}$, and $d_{XY}$ in windows across the genome (available here: https://github.com/bnavarrodominguez/sd_popgen; ***Navarro-Dominguez, 2022b***, copy archived at swh:1:rev:e012c1df579871600334847e254a1ecc6c053592). To calculate $F_{ST}$ values, we used the Weir-Cockerham estimator (***Weir and Cockerham, 1984***). Only those sites with a minimum sample depth of 8 were included in the $F_{ST}$ and Tajima's $D$ calculations. We determined window size by the number of 'acceptable sample depth' sites (and not, for example, a particular range of chromosome coordinates). As Tajima's $D$ is sensitive to the numbers of segregating sites in a sample (***Schaeffer, 2002***), we also estimated $D/D_{min}$; that is, the ratio of Tajima's $D$ to its theoretical minimum ($D_{min}$) when all sites in a window are singletons (***Tajima, 1989***; ***Figure 2—figure supplement 2***). To confirm that repeats were not interfering with our results, we ran our population genomics pipeline masking SNPs in repetitive elements identified by RepeatMasker (***Smit et al., 2013***), which yielded equivalent results (***Supplementary file 6***, Sheet 1).

## Age of the sweep

We calculated overall $S_{In(2R)Mal}$, $\pi_{In(2R)Mal}$ and Tajima's $D_{In(2R)Mal}$ from the *SD-Mal* SNP set using our same Perl script (available here: https://github.com/bnavarrodominguez/sd_popgen), using a single window of 9.5 Mb within the boundaries of *In(2R)Mal*. To account for gene conversion, we calculated an additional set of summary statistics masking the SNPs annotated as shared by at least one of the *SD+* libraries. We estimated the time since the most recent selective sweep using an ABC method with rejection sampling. We modeled the selective sweep as an absolute bottleneck ($N_t = 1$) at some time ($t$, $4N_e$ generations) in the past. We performed simulations in *ms* (***Hudson, 2002***), considering a sample size of 9 and assuming no recombination in the ~9.92 Mb of *In(2R)Mal*. To enrich for neutral mutations, we only considered non-coding SNPs in intergenic and intronic regions. We simulated with values of $S_{Sim}$ drawn from a uniform distribution ±5% of $S_{In(2R)Mal}$. We considered a prior uniform distribution of time of the sweep ($t$) ranging from 0 to $4N_e$ generations, that is, 0–185,836 years ago, considering *D. melanogaster* $N_e$ in Zambia 3,160,475 (***Kapopoulou et al., 2018***), frequency of *In(2R) Mal* 1.47% and 10 generations per year. The rejection sampling algorithm is as follows: (1) draw $S_{Sim}$ and $t$ from prior distributions; (2) simulate 1000 samples using the coalescent under a selective sweep model; (3) calculate average summary statistics for drawn $S_{Sim}$ and $t$; (4) accept or reject chosen parameter values conditional on $|\pi_{In(2R)Mal} - \pi_{Sim}| \le \varepsilon$, $|D_{In(2R)Mal} - D_{Sim}| \le \varepsilon$; (5) return to step 1 and continue simulations until $m$ desired samples from the joint posterior probability distribution are collected. For estimates of $t$, $\varepsilon$ was set to 5% of the observed values of the summary statistics (in step 4) and $m$ was set to 10,000. These simulations were performed with parameters calculated using all the SNPs in non-coding regions of *In(2R)Mal* and also excluding SNPs shared with *SD+* chromosomes to account for gene conversion. We simulated 100,000 samples with the resulting estimated $t$ and $S_{In(2R)Mal}$, under our sweep model, under a constant size population model, and under an exponential growth model.

For the growth model, we assumed an exponential growth rate of $\alpha = 0.26$, based on parameters estimated for Zambia (exponential growth from past $N_e = 1{,}137{,}712$ to present $N_e = 3{,}160{,}475$ in the last 72,005 years) (*Kapopoulou et al., 2018*), scaling $N_e$ by the frequency of *SD-Mal* in Zambia (1.47%). We calculated two-sided p-values for $\pi$ and Tajima's $D$ using an empirical cumulative probability function (ecdf) in R (*R Development Core Team, 2019*). We estimated the maximum a posteriori estimate as the posterior mode and 95% credibility intervals (CIs) in R (*R Development Core Team, 2019*).

## Recombination

For estimates of recombination, we filtered the SNPs in *In(2R)Mal* to variable positions genotyped in all of the nine *ZI-SD* samples and excluded singletons, resulting in a total of 338 SNPs. We estimated pairwise linkage disequilibrium ($r^2$) using PLINK v1.9 (*Purcell et al., 2007*). We discarded $r^2$ data calculated for pairs of SNPs flanking the internal *In(2R)Mal* breakpoints. For comparison, we estimated pairwise linkage disequilibrium in the same region of *In(2R)Mal* for $SD^+$ uninverted *2R* chromosome arms and, for comparison, in $SD^+$ *In(2R)NS* inversion and in $SD^+$ *In(2L)t* inversions. For $SD^+$ chromosomes, we applied the same filters (variable, non-singleton SNPs), plus an SNP 'thinning' to 1 SNPs/kb to get a manageable set of results. To investigate the possibility of crossing over between *SD-Mal* chromosomes, we used RecMin (*Myers and Griffiths, 2003*) to estimate the minimum number of crossovers between the 338 biallelic, non-singleton SNPs in *In(2R)Mal*. RecMin input is a binary file, which we generated using *SD-ZI125* as an arbitrary reference for *SD*, assigning 0 or 1 on each position depending on if it was the same base or different. Maximum likelihood trees to establish relationships between *SD-Mal* haplotypes based on these 338 SNPs were estimated using RAxML-NG (*Kozlov et al., 2019*).

Runs of shared and private SNPs were identified in R, using all SNPs (including singletons). A run of SNPs is defined as a region from 5′ to 3′ where all the SNPs are in the same category (shared or private). Distance between the first and the last SNP of a category is considered length of the run. The region between the last SNP of a category and the first SNP of the alternative category is considered distance between runs. Because our sample size is small, we may underestimate the number of shared SNPs, as some private SNPs may be shared with some $SD^+$ chromosomes that we have not sampled.

## TE calling

We used a TE library containing consensus sequences of *Drosophila* TE families (*Chang and Larracuente, 2019*). With this library, we ran RepeatMasker (*Smit et al., 2013*) to annotate reference TEs in the *D. melanogaster* (BDGP6) genome. To detect genotype-specific TE insertions in our Illumina libraries, we used the McClintock pipeline (*Nelson et al., 2017*), which runs six different programs with different strategies for TE calling. We collected the redundant outputs from RetroSeq (*Keane et al., 2013*), PoPoolationTE (*Kofler et al., 2012*), ngs_te_mapper (*Linheiro et al., 2012*), TE-Locate (*Platzer et al., 2012*), and TEMP (*Zhuang et al., 2014*), discarded the calls produced by TEMP based on non-evidence of absence, and then merged the insertions detected by all different programs, considering the same insertion those of the same TE closer than a distance of ±600 bp, as described in *Bast et al., 2019*. To reduce false positives, we only considered TE insertion calls that were predicted by more than one of the methods. To account for differences in library read number and/or length between datasets, we report the TE counts for *2R* normalized by the TE count for chromosome *2L* for the same library (*Figure 6B*). To assess whether library differences qualitatively affect our results, we repeated the above TE analysis on a set of 3 million randomly selected paired-end reads, trimmed to a fixed length of 75 bp, from each library and report TE count for chromosomes *2R* and *2L* separately (*Figure 6—figure supplement 3*).

## Acknowledgements

This work was funded by the National Institutes of Health (NIH), National Institute of General Medical Sciences (R35GM119515 and NIH-NRSA F32GM105317 to AML), Stephen Biggar and Elisabeth Asaro fellowship in Data Science to AML, a David and Lucile Packard Foundation grant and University of Rochester funds to DCP. We thank Dr Danna Eickbush for assistance with genomic DNA preparation for nanopore sequencing. We also thank the University of Rochester CIRC for access to computing cluster resources and UR Genomics Research Center for the library construction and sequencing.

## Additional information

### Funding

| Funder | Grant reference number | Author |
|---|---|---|
| National Institute of General Medical Sciences | R35 GM119515 | Amanda M Larracuente |
| Stephen Biggar and Elisabeth Asaro Fellowship in Data Science | | Amanda M Larracuente |
| David and Lucile Packard Foundation | | Daven C Presgraves |
| University of Rochester | | Daven C Presgraves |
| National Institute of General Medical Sciences | F32GM105317 | Amanda M Larracuente |

The funders had no role in study design, data collection and interpretation, or the decision to submit the work for publication.

### Author contributions

Beatriz Navarro-Dominguez, Formal analysis, Investigation, Methodology, Software, Validation, Visualization, Writing – original draft, Writing – review and editing; Ching-Ho Chang, Formal analysis, Investigation, Software, Writing – review and editing; Cara L Brand, Investigation, Writing – review and editing; Christina A Muirhead, Investigation, Methodology, Software, Writing – review and editing; Daven C Presgraves, Conceptualization, Formal analysis, Funding acquisition, Investigation, Project administration, Resources, Supervision, Writing – original draft, Writing – review and editing; Amanda M Larracuente, Conceptualization, Data curation, Formal analysis, Funding acquisition, Investigation, Methodology, Project administration, Software, Supervision, Writing – original draft, Writing – review and editing

### Author ORCIDs

Beatriz Navarro-Dominguez http://orcid.org/0000-0003-4077-8696
Ching-Ho Chang http://orcid.org/0000-0001-9361-1190
Daven C Presgraves http://orcid.org/0000-0003-4337-4072
Amanda M Larracuente http://orcid.org/0000-0001-5944-5686

### Decision letter and Author response

Decision letter https://doi.org/10.7554/eLife.78981.sa1
Author response https://doi.org/10.7554/eLife.78981.sa2

## Additional files

### Supplementary files

• Supplementary file 1. Illumina data used in this paper, including those from *Lack et al., 2016*. List of NCBI SRA accession numbers, genotype, number of reads, and estimated coverage for each line (Sheet 1). Per-chromosome depth of the *SD-Mal* lines sequenced in this paper (Sheet 2).

• Supplementary file 2. Measures of drive strength of *+,+,px* chromosomes in *b c px/b c px* x *+ + px/b c px* crosses (see *Figure 1—figure supplement 4*): number of male and female *b c px* and *+ + px* progeny; *n*, total progeny; *k*, average proportion of progeny inheriting the *+ + px* chromosome; *w*, relative viability of *+ + px* chromosomes; *k\**, average proportion of progeny inheriting the *+ + px* chromosome, corrected for viability. Genotypes (where *BL156* or *SD* means consistent with either of those parental alleles): *Sd* (*2L*:19.44); inferred from visible markers and double checked with primers in *Presgraves et al., 2009*; *In(2R)Mal* distal breakpoint (*2R*:18.77; inferred from visible markers and double checked with primers in *Supplementary file 4*), *sano* (*2R*:18.87; inferred from molecular markers, primers in the Materials and methods section); *CG15666* (*2R*:21.34; inferred from molecular markers, primers in the Materials and methods section) and *px* (*2R*:22.49; inferred from visible markers).

- Supplementary file 3. Number and density (SNPs/Mb) of private and shared SNPs in *In(2R)Mal-p*, *In(2R)Mal-d* and their overlapping area (inverted twice).
- Supplementary file 4. Sequence and coordinates of primers used to validate *In(2R)Mal* breakpoints.
- Supplementary file 5. Results of screening for the *Sd-RanGAP* duplication, *In(2R)Mal*, *In(2R)NS*, and *In(2L)t* inversions in DPGP3 dataset (*Lack et al., 2016*), and comparison with the most recent inversion calls (http://johnpool.net/Updated_Inversions.xls, last accessed 07/13/2020).
- Supplementary file 6. Average nucleotide diversity ($\pi$) per nucleotide and empirical standard deviation estimated in 10 kb windows along chromosome 2, for $SD^+$, $SD$-$Mal$ and $SD^+$ scaled by the estimated frequency of $SD$-$Mal$ chromosomes ($SD^+ \times f$; $f = 1.47\%$); with repetitive elements masked (Sheet 1) and excluding $SD$-$ZI138$ (Sheet 2).
- MDAR checklist

## Data availability

Raw sequence data are deposited in NCBI's short read archive under project accession PRJNA649752. All code for data analysis and figure generation is available in Github (https://github.com/bnavarro-dominguez/sd_popgen, copy archived at swh:1:rev:e012c1df579871600334847e254a1ecc6c053592). Data and code will be deposited in Dryad digital repository.

The following dataset was generated:

| Author(s) | Year | Dataset title | Dataset URL | Database and Identifier |
|---|---|---|---|---|
| Navarro-Dominguez B, Chang C, Brand C, Muirhead C, Presgraves D, Larracuente AM | 2022 | Epistatic selection on a selfish Segregation Distorter supergene: drive, recombination, and genetic load | https://doi.org/10.5061/dryad.4qrfj6qch | Dryad Digital Repository, 10.5061/dryad.4qrfj6qch |

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
