## [Editor Report]

The work advances our understanding of the *Segregation Distorter* (*SD*) complex in *Drosophila melanogaster*. *SD*, the classic example of a selfish chromosome, consists of two tightly linked genetic elements and thus qualifies as a supergene. The work also excels through particularly careful analyses.

---

## [Decision Letter]

[Editors' note: this paper was reviewed by Review Commons.]

---

## [Author Response]

Reviewer #1 (Evidence, reproducibility and clarity (Required)):This study examines the Segregation Distorter (SD) polymorphism in a single population in Zambia, Africa. The goals for this study are: (1) to identify the structural features of the SD-Mal haplotype, including the organization of the insensitive Rsp allele and the In(2R)Mal rearrangements; (2) to characterize the genetic function of In(2R)Mal and its role in drive; (3) to infer the population genetic history of the rapid rise in frequency of SD-Mal in Zambia; and (4) to explore the evolutionary consequences of reduced recombination on SD-Mal haplotypes. To accomplish these goals, Nanopore long read sequencing was used to generate a de novo assembly of one SD-Mal haplotype followed by Illumina short read resequencing of nine driving SD haplotypes from a single population in Zambia. The long read assembly of the SD-Mal with the overlapping inversions was used to infer that the distal inversion happened first followed by the proximal inversion.Recombinants were generated with crosses between the SD-Mal and a multiply marked strain show that the two inversions are required for SD drive. This is interesting because the two inversions do not include either SD or Rsp implying an enhancer of drive is located within the inverted region. The paper uses the decay of LD between the SD locus and the inversions to show that LD should decay rapidly suggesting that the association of SD and the inversions is maintained by epistatic selection (see FULLER et al. 2020 for a similar argument for Sex-ratio in D. pseudoobscura).The paper used population genomic analysis of SNPs to show that nucleotide heterozygosity is reduced acros 25 Mb of the SD, Rsp, and Inversion chromosome suggesting a recent origin of the SD-Mal chromosome.The paper used analysis of synonymous and non-synonymous ( S and N) SNPs to infer the degree of deleterious mutations in SD-Mal. The reasoning is that most nonsynonymous SNPs reflect deleterious alleles. The analysis presented here examines the ratio of N/S as a proxy for genetic load. The inverted chromosome has a higher N/S than non-inverted arrangements suggesting that there is higher genetic load or deleterious mutations on SD-Mal chromosomes explaining homozygous lethality of SD-Mals. One question that is not answered by this analysis is what is the frequency of the non-synonymous mutations. How often would they be allelic such that different SD-Mals would be homozygous lethal? Or would only self-SD-Mals be homozygous lethal?

Indeed, most combinations of SD Mal chromosomes are viable and fertile in complementation tests (see Presgraves et al. 2009 and Brand et al. 2015), suggesting that most of the deleterious mutations causing lethality or sterility are unique to each haplotype.

TEs are found in much greater numbers in SD-Mal chromosomes consistent with the idea that low recombination regions are a sink for the accumulation of TEs.This study examines the Segregation Distorter (SD) polymorphism in a single population in Zambia, Africa. SD is a meiotic drive element that is composed of two tightly linked parts, a driver locus Ran-Gap and the driver's target Responder (Rsp). Ran-Gap is duplicated on the SD chromosome while Rsp is a satellite DNA with SD sensitive and insensitive alleles. The sensitive Rsp allele is a larger satellite that the Rsp insensitive allele. The SD driver allele is linked to a Rsp insensitive allele and targets Rspsensitive chromosomes limiting their transmission during meiosis. This paper maps the location of breakpoints of two inversions that are associated with the SD chromosome and increase the transmission of the SD chromosome. The paper determines the evolutionary sequence of the two inversion events. Evolutionary genomic analysis demonstrates that the SD-Mal chromosome is recent and that the linked inversions enhance the SD phenotype. This is interesting because the distance between the inversions and the two critical loci for the SD phenomenon is sufficient to be randomly shuffled. The fact that they are nonrandomly associated supports the hypothesis that at least one SD enhancing locus is found within the inversions and that strong epistatic selection is maintaining the association.

Thank you for this nice summary. We respond to your comments point-by-point below.

No additional experiments are needed.The data and methods are presented in such a way that they are reproducible. The data analyses are appropriateMajor Comments1. Page 7 lines 159-176. The issue at play here is the position effect where the inversion breakpoints create variation that could play a role in selection for the phenotype. The paper describes how the breakpoints have disrupted some genes including one that is highly expressed in testes. Another possibility that should be mentioned is that the breakpoints could disrupt gene expression of neighboring genes either because of shuffling regulatory sequences or altering topologically associated domains (TADs). Hou et al. (2012) and others have developed TAD maps for *D. melanogaster*. Do the inversion breakpoints disrupt TADs potentially leading to altered gene expression that could create selectable variation. I agree that future work will be needed to consider all these possibilities.

This is an interesting possibility. Thank you for this suggestion. All four inversion breakpoints disrupt physical domains reported in Hou et al. (2012). However, TADs can vary between tissues, and some studies show that TAD reorganization by inversions does not impact gene expression in *Drosophila* (Ghavi-Helm et al. 2019 ; Meadows et al. 2010). As Torosin et al. 2020 propose, there might be functionally distinct TAD subtypes, some of them highly conserved and some of them might tolerate rearrangements and evolve rapidly. Thus, it is hard to know the impact of this disruption on SD-Mal. We therefore prefer to be cautious. We revised the text to say: “Even for genes that are not directly interrupted by the inversion breakpoints, the chromosomal rearrangements may disrupt the regulation of nearby genes if, e.g., they affect the organization of topologically associating domains (TADs; reviewed in (SPIELMANN et al. 2018)). The In(2R)Mal inversion breakpoints disrupt physical domains reported in Hou et al. (HOU et al. 2012), however inversion-mediated disruptions of TAD boundaries do not necessarily affect gene expression (GHAVI-HELM et al. 2019). Future work is required to determine if the inversions affect gene expression near the breakpoints and if CG10931 has a role in the SD-Mal drive phenotype.” (lines 194-199)

2. Page 8 Lines 207-208. Can more detail be provided about how these recombinants were infer from molecular markers? Is there any evidence for gene conversion events with non-SD chromosomes, i.e., 100 to 200 bp segments? See Rozas and Aguade (ROZAS AND AGUADÉ 1993) and Betran et al. (1997).

Thank you for noting that details were missing about how we genotyped based on molecular markers. We amplified a region from the genes sano (2R: 18.87) and CG15666 (2R: 21.34) using primers that yield amplicons of different sizes in SD^+^ and SD-Mal chromosomes. We now include the PCR primers and a description of our approach in the Methods (lines 583-587).

We infer gene conversion events with non-SD chromosomes from the pattern of shared and private SNPs (see the section “Recombination on SD-Mal supergenes”, starting at line 386).

3. Page 12 Lines 269-279. Tajima's D is sensitive to the numbers of segregating sites within a window (SCHAEFFER 2002). One way to normalize D is to use the ratio of D to its theoretical minimum Dmin when all sites are singletons (see equations 43-45 in TAJIMA 1989). For an example, see Figure 5 in (FULLER et al. 2017).

Thank you for this suggestion. We estimated Tajima’s D normalized by Dmin and now we show the results in a new supplementary figure (Figure S9). The results are very similar to Tajima’s D without normalization and therefore our conclusion is the same. We elected to keep the standard Tajima’s D in the main text (Figure 2) and present the normalized D in the supplement (Figure S9).

4. Page 14 Lines 304-306. I would have initially agreed with the paper that shared SNPs between SD-Mal and the ancestral arrangement reflect gene conversion. Using SLIM3 (HALLER AND MESSER 2019) to simulate the origin of new inversions, there is a period of time soon after the origin of the new arrangement where SNPs are polymorphic in the ancestral arrangement and one of the ancestral alleles is fixed in the derived arrangement. Would this site be removed as evidence of gene conversion or would both arrangements need to be segregating for both variants?

To detect evidence of gene conversion, we only consider sites that are polymorphic in In(2R)Mal and so we would not consider sites that are fixed in In(2R)Mal.

5. Page 16 Lines 347-366. Recombination in the form of gene conversion would not only happen between SD-Mal chromosomes, but more likely to occur in pairings with non-SD chromosomes.

Yes, we agree. However this section is discussing crossing over. The discussion of gene conversion is in the next paragraph (starting at line 416). To make it clearer that here we are referring to crossing over, we added “Therefore, recombination via crossing over may occur between SD-Mal chromosomes in SDi/SDj heterozygous females.” (line 390)

6. Figure 1b. More explanation is needed to explain what is being shown in the lower part of this figure. Does a black line indicate presence of a repeat or TE? An explanation should be added to the figure legend.

Yes, the black line indicates the presence of a repetitive element. We adjusted the legend to make this clearer: “The tracks below indicate the presence of types of repetitive elements found at this locus. Black lines indicate the presence of a repeat type in the reference genome. Grey shading shows where Rsp repeats are in the reference genome.”. Thanks for the suggestion.

7. Figure 1c. It might be useful for readers that don't think about overlapping inversions often to provide schematics for the two possible rearrangement scenarios to help understand the inference about why the distal inversion had to happen first. The two possible scenarios are: 1. distal first proximal second leads to red, blue, yellow, green, purple versus 2. proximal first distal second leads to red, green, blue, yellow, purple. The first scenario fits the data. What does the striping of the block mean?

Thanks very much for the suggestion to add schematics. We added a supplemental figure (supplemental figure S2) showing schematics for the different scenarios. The apparent striping of the blocks reflects the similarity profile estimated by Mauve: the height corresponds to the sequence conservation and is calculated based on the alignment column entropy in an aligned region. We now include a brief description in the legend: “The colored rectangles correspond to locally collinear blocks of sequence with the height of lines within the block corresponding to average sequence conservation in the aligned region (Darling et al. 2010).”

Literature CitedBetran, E., J. Rozas, A. Navarro and A. Barbadilla, 1997 The estimation of the number and the length distribution of gene conversion tracts from population DNA sequence data. Genetics 146: 89-99.Fuller, Z. L., G. D. Haynes, S. Richards and S. W. Schaeffer, 2017 Genomics of natural populations: Evolutionary forces that establish and maintain gene arrangements in Drosophila pseudoobscura.Molecular Ecology 26: 6539-6562.Fuller, Z. L., S. A. Koury, C. J. Leonard, R. E. Young, K. Ikegami et al., 2020 Extensive Recombination Suppression and Epistatic Selection Causes Chromosome-Wide Differentiation of a Selfish Sex Chromosome in Drosophila pseudoobscura. Genetics 216: 205-226.Haller, B. C., and P. W. Messer, 2019 SLiM 3: Forward Genetic Simulations Beyond the Wright-Fisher Model. Molecular Biology and Evolution 36: 632-637.Hou, C., L. Li, Zhaohui S. Qin and Victor G. Corces, 2012 Gene density, transcription, and insulators contribute to the partition of the Drosophila genome into physical domains. Molecular Cell 48: 471-484. Rozas, J., and M. Aguadé, 1993 Transfer of genetic information in the rp49 region of Drosophila subobscura between different gene arrangements. Proceedings of the National Academy of Sciences USA 90: 8083-8087.Schaeffer, S. W., 2002 Molecular population genetics of sequence length diversity in the Adh region of Drosophila pseudoobscura. Genetical Research 80: 163-175.Tajima, F., 1989 Statistical method for testing the neutral mutation hypothesis by DNA polymorphism. Genetics 123: 585-595.Reviewer #1 (Significance (Required)):This study examines the Segregation Distorter (SD) polymorphism in a single population in Zambia, Africa. SD is a meiotic drive element that is composed of two tightly linked parts, a driver locus Ran-Gap and the driver's target Responder (Rsp). Ran-Gap is duplicated on the SD chromosome while Rsp is a satellite DNA with SD sensitive and insensitive alleles. The sensitive Rsp allele is a larger satellite that the Rsp insensitive allele. The SD driver allele is linked to a Rsp insensitive allele and targets Rspsensitive chromosomes limiting their transmission during meiosis. This paper maps the location of breakpoints of two inversions that are associated with the SD chromosome and increase the transmission of the SD chromosome. The paper determines the evolutionary sequence of the two inversion events. Evolutionary genomic analysis demonstrates that the SD-Mal chromosome is recent and that the linked inversions enhance the SD phenotype. This is interesting because the distance between the inversions and the two critical loci for the SD phenomenon is sufficient to be randomly shuffled. The fact that they are nonrandomly associated supports the hypothesis that at least one SD enhancing locus is found within the inversions and that strong epistatic selection is maintaining the association. This work would appeal to audiences in the evolutionary genomics field who are interested in genome rearrangement related to phenotypic variation.My expertise is in evolutionary genomics and I have done similar work on a meiotic drive system on X chromosomes that lead to altered Sex ratios. The Sex ratio system also involves inversion mutations associated with the Sex ratio phenotype.Referees cross-commenting sessionReviewer 1I think the composite set of reviews all agree about the importance of this study. One of few studies that has sequenced a driving chromosome and provided some insightful conclusions. Each of the reviewers touch on slightly different issues, but the set of comments will definitely improve the paper. I think the paper strikes the right balance of addressing important issues without going into too much detail that might hide the interesting results.Reviewer 3I agree with reviewer 1.Reviewer 2I also agree – thank you for the nice summary!Reviewer #2 (Evidence, reproducibility and clarity (Required)):SD is a classic example of an autosomal segregation distorter. By combining genetic analyses, long read dna sequencing of one haplotype, and population genomics data, this manuscript provides the first description of the sequence of this selfish element, confirming several previous observations (e.g. general structure), and shedding new light on its history and on the selective pressures that have shaped it (a recent selective sweep driven by epistemically selection, and the accumulation of an excess of nonsynonymous likely due do a reduction in the efficacy of purifying selection). The functional role of different parts of the element is also tested by looking at the distortion capacity of SD/Wild type recombinants.I really enjoyed how thorough the analyses provided here are, and found the conclusions very convincing. My only difficulty was that because so much work is presented here, there was sometimes not much space to put results fully into context. For instance, how the age of the sweep is determined is explained in detail, but the importance of the number is only briefly alluded to in the conclusions. The results could perhaps also be discussed more in the context of what has been found for other drivers (e.g. the excess of nonsynonymous SNPs).

Thanks very much for your positive evaluation and for your suggestion. We added some additional context in the results (starting at line 347 and additional context referenced below) and respond to your other comments point-by-point below.

Other comments:– Can the simulations distinguish between SDmal sweeping across a stable SD population (i.e. without changing the 1.4% frequency) versus sweeping in a way that increased SD frequency in the population? It would have been helpful to put these results in the context of what is known about frequencies of SD, and what selective pressures shape them.

This is an interesting point. We are unsure of the detailed population dynamics of driving chromosomes. It is possible that SD-Mal could have increased in population frequency following the sweep before returning to the lower frequencies that we observe. We do not simulate this scenario but instead assume that following the replacement event SD-Mal rose to the same frequency as the haplotype that it replaced (similar to the scenario considered by Charlesworth and Hartl 1978). This assumption is in part motivated by an observation that Rayla Temin and Kreber made in Wisconsin, USA. Based on longitudinal data collected ~25 years apart in the same location, they observed that a SD72 haplotype replaced the previously common SD5 haplotype without a major change in the overall SD population frequency (~3%; Temin and Kreber 1981). Given the short timescale of that study, we do not consider a possible transient rise in frequency. Additionally, the selective sweep complicates any inference by coalescence, because we have no history for anything before the sweep. We added text in the Introduction to provide additional context (lines 86-87): “Multiple factors likely contribute to the low frequency of SD in populations: negative selection, insensitive Rsp alleles, and unlinked suppressors (reviewed in LARRACUENTE AND PRESGRAVES 2012). Two independent longitudinal studies suggest that SD haplotypes can replace each other in populations over short time scales (<30 years) (TEMIN AND KREBER 1981; BRAND et al. 2015) without major changes in the overall population frequency of SD (TEMIN AND KREBER 1981).”

– Some runs of shared snps seem to be in longer blocs – can you really exclude double crossovers happening with SD+ (as claimed in the text, e.g. l. 105)?

We cannot exclude double crossovers, but at the same time we have little evidence for them. Double crossover events are relatively rare and are widely spaced, with an average distance between them of 10.5 Mb (Miller et al. 2016). The longest block of shared SNPs we found is 50.2kb, which is still much shorter than what is expected by double crossover (e.g. the smallest double crossover event detected in Miller et al. 2016 is 1.5Mb). We also did not find any long regions standing out as having a similar nucleotide diversity as SD^+^ chromosomes. One explanation for these longer tracts is that runs of shared SNPs can be sparse in a long block from multiple gene conversion events, and uninterrupted by private SNPs in the middle because of the young age of the inversion. We updated the text to acknowledge that we cannot rule out double crossovers (lines 423-425): “Although we cannot exclude the contribution of double crossovers, we note that 62.2% (89 out of 143) of the shared SNP runs are <1 kb, 80.4% (115 out of 143) are <10 kb (Figure 5b), and the longest run is ~50.2 kb. These sizes are more consistent with current estimates of gene conversion tract lengths in *D. melanogaster* than with double cross-over (COMERON et al. 2012; MILLER et al. 2016).”

– Figure 1 was a bit hard to read– Panel a has no labels on the axes, and the numbers are very smallThe letters denoting the markers on panel c look exactly the same as the panel letters -- could capital letters be used for the panels?– Maybe the box with the syntenic blocs could be its own panel? It would make it easier for readers to check in the legend what it is showing.– What do the lines joining the syntenic blocs mean? At first I assumed they would be showing breakpoints, as that would seem like the logical thing to show.

Thank you for the suggestions. We increased label font size and now use capital letters for panels to make the figures clearer. We also made the synteny-block figure its own panel. The lines in this plot join syntenic blocks so that the reader can see where the block is the alternative arrangement. Breakpoints are between blocks.

– L. 182 where is the data for the reduction in recombination coming from? No figure or table is cited.

We had reported the reduction in genetic distance in the text instead of providing a table. We now include these data, including the number of recombinants, in a new Table 1.

– The paragraph starting at l. 178 starts with the known observation that the inversion is essential for SDmal drive, and ends with essentially the same conclusion. Maybe it could be rephrased?

Thank you for this feedback. We knew from previous work that, in natural populations from Africa, SD chromosomes sampled bearing both Sd and In(2R)Mal show very strong drive, but SD chromosomes bearing just Sd could not drive at all (Presgraves et al. 2009, Brand et al. 2015). However, we did not have direct evidence that recombination between Sd and In(2R)Mal on a driving SD chromosome creates Sd and In(2R)Mal recombinants where neither drive and are instead inherited in Mendelian proportions, as we show in Table 2. We edited the text to make this point clearer (paragraph starting at line 202).

– Table 1: it could just be me, but I found the recombinants and their markers hard to make sense of. Maybe a visual schematic could be provided (as a supplementary figure at least) for readers with no experience in fly genetics?

Thank you for the suggestion. We now include a new supplemental figure showing the crossing scheme and diagram of possible genotypes (supplemental figure S4)

– Table 2: I wonder if this could be simplified and the expectations (outside of linked region, SDmal = SD+; inside, SDmal =< f*SD+) explicitly stated in the text. (This is done for SDmal but not for the adjacent regions). I found it a bit confusing to have comparisons for whole chromosomal arms when the expectation is a mix of both.

Thank you for the suggestion. We modified this table (now Table 3) to remove whole chromosome arms and added further explanation in the Table caption. We also added a brief sentence in the text (lines 286-288) “While nucleotide diversity outside of the SD-Mal supergene region is comparable to SD^+^ (Table 3 row 1), diversity in the supergene region is significantly lower than expected even when scaled by its frequency (Table 3 row 4), suggesting that the low frequency of SD-Mal cannot fully explain its reduced diversity.”

Reviewer #2 (Significance (Required)):Although supergenes are in fashion, there are still only a few examples of fully sequenced meiotic drivers, and the addition of SD is an important contribution. The combination of methods and detailed analysis is also novel and crucial for understanding the evolutionary pressures shaping drivers.Reviewer #3 (Evidence, reproducibility and clarity (Required)):Overall, this is an excellent paper, which presents new and illuminating results on the classic system of meiotic drive, the Segregation Distorter complex of *Drosophila melanogaster*. It is a nice combination of genetics and population genomics, and is clearly written and interesting to read. I have not, however, evaluated the sequencing and bioinformatic details, as these are beyond my expertise.

Thank you for your positive feedback and thoughtful suggestions. We respond to your comments point-by-point below.

I have a number of relatively minor comments that I hope will help to improve the paper.

l.42 Inversions were discussed in Charlesworth and Hartl 1978.

Done

l.43 I think Turner 1977 (Turner, J. R. G. 1977. Evol. Biol. 10:163-206.) is a better reference; the 1967 paper does not deal with recombination modifiers like inversions.

Done

l.54 These are rather out of date references on Hill-Robertson effects, etc. The field has moved on a lot since then, especially the discovery of the interference selection limit in very low recombination regions (e.g. Charlesworth, B. et al. 2010. Genetic recombination and molecular evolution. Cold Spring Harb. Symp. Quant. Biol. 74:177-186).

Thank you for the suggestion. We updated these references

l.55-57 This seems a bit garbled: delete ', they' on l.57, perhaps.

Done. We revised to: “The degeneration of drive haplotypes is not inevitable, however. Different drive haplotypes that complement one another may be able to recombine, if only among themselves (DOD et al. 2003; PRESGRAVES et al. 2009; BRAND et al. 2015).” (lines 56-58).

l.60-61 Maybe mention selection for loosely linked suppressors of drive.

We do not think that this is a natural place to bring up suppressors of drive at this point, so we did not mention it here.

l.74 Maybe mention the problem that pericentric inversions can have problems with crossing over- Coyne, JA et al. 1993 Genetics 134: 487-496 have relevant data that suggest how this can be mitigated.

We modified the sentence to add: “In heterozygotes with a pericentric inversion, recombination in the inverted region generates aneuploids and therefore reduced fertility, although this effect might be mitigated by strong suppression of recombination (Coyne, 1993).“ (lines 75-77)

l.86 It would be helpful to mention the fraction of the chromosome arm that is covered by the inversions.

Thanks for the suggestion. We modified this sentence to say: “SD-Mal has a pair of rare, African-endemic, overlapping paracentric inversions spanning ~40% of 2R”. (lines 92-94)

l.123 It's not clear what Hs2st in the figure refers to.

We added in the figure legend: “The gene Hs2st occurs in the first intron of RanGAP, and it is also duplicated in the Sd locus (Hs2st-2). “

l.165 This order looks correct, but maybe a diagram to show this would be helpful (e.g. in the SI).

Thank you for the suggestion. We have added a schematic showing the possible order of inversions as a supplemental figure (figure S2) and this line to the text (lines 185-186): “Note that any rearrangement different than distal first, proximal second, leads to a different outcome (Suppl. Figure S2).”

l.183 In the methods (l.522), they mention the standard multiply marked 2nd chromosome al, dp, b, pr, c, sp; why were only b, c and px used?

The locations of al (2L: 0.34), dpy (2L: 4.44) and speck (2R: 24.14) were not useful for our goal of studying recombinants within the SD-Mal supergene, since they are further from the supergene than the markers we used: b (2L:13.82) and px (2R:22.49). While pr (2L:20.07) might have been useful, the phenotype (purple eyes) was difficult to distinguish in the crosses, since our stock of SD-Mal flies has white eyes. We edited the methods to say: “To estimate recombination frequencies and obtain SD recombinant genotypes, we used a stock (al[1] dpy[ov1] b[1] pr[1] c[1] px[1] speck[1], BDSC156, Bloomington *Drosophila* Stock Center), which has three visible, recessive markers on chromosome 2 that are informative about recombinants involving SD: black (b, 2L: 13.82), curved (c, 2R:15.9) and plexus (px, 2R:22.49). Our SD flies are white-eyed, therefore we did not score purple (pr, an eye color phenotype).”

l.207 Where are the results for the molecular markers shown?

We added a supplementary table with the results for molecular markers (now Suppl. Table S2)

l.221 What is the confidence interval on this? Maybe also give the correlation coefficient as well.

We now report the correlation coefficient. We are unclear about what confidence interval requested here, but we have assumed that you would like to account for stochasticity in the rate of decay of LD due to the interaction of genetic drift and recombination. Unfortunately, we are unaware of (and were unable to identify) population genetic theory solutions for the variance in the expected decay of LD across generations. However, we suggest that the use of the expected deterministic decay in LD in our simple calculation is reasonably justified by the very, very large population recombination parameter for Sd and the proximal breakpoint of In(2R)Mal (Ne*r > 10^4^). In particular, we assume that N_e_ = 10^6^ and use an estimated, sex-averaged recombination frequency of r = 0.025, giving N_e_*r = 25,000.

l.235 There is a problem with the s.e.'s on the diversity values; these SNPs are not independent of each other, as the LD plot shows. One option would be to treat them as part of a non-recombining chunk of genome, and use the standard coalescent formula for their variance to get the s.e.'s. This would be conservative.

For the non-recombining region, we now report the standard deviation from the variance in Pi, using the formula B5.6.3 from Charlesworth and Charlesworth (2010) pp. 212-213 in Table 2.

l.250 'seem' is better than 'be'. They might want to look at this paper, which models the spread of a mutation to a balanced state: Zeng, K. et al. 2021. Genetics 218: iyab055.

We see your point. However, as the prose construction ‘may not seem surprising’ is a bit awkward, we have opted for an equivalent alternative phrasing: “The reduced nucleotide diversity among SD-Mal might be expected given its low frequency in natural populations.” This is an interesting paper but we did not come up with a natural way to tie this into our text.

l.266 It would be good to explain that the low frequency of the inversion means that it is largely carried in heterozygotes, where crossing over (but probably not gene conversion) is suppressed.

We mention on line 296 that there is a low recombination frequency between SD and SD+. We also mention that crossing over is suppressed in SD/SD+ heterozygotes on line 417.

l.273-275 The use of Fst needs more explanation, since it is usually used for between-population comparisons. Here, you are treating inversion and standard arrangements as two subpopulations. It may also be worth mentioning that the Fst statistic is another way of describing the LD between SNPs and the inversion (see the Zeng et al. paper). Also the high Fst and low dxy is consistent with the recent sweep and low diversity within the inversion (see Charlesworth, B/ 1998 MBE 15:538-543; Cruickshank, T. and Hahn, M. 2014. Mol Ecol 23:3133-3157).

We added an additional explanation for the use of FST (line 298) and now mention explicitly that FST and DXY are also consistent with the recent sweep within SD-Mal (line 303-305).

However, FST is not another way of describing the LD between SNPs and the inversion. FST is a measure of population differentiation due to genetic structure. It takes into account the variance of frequencies of the same allele between two populations, and the probability of identity by descent. Linkage disequilibrium estimates non-random association of alleles at different loci, in the same chromosome, in a given population. In this particular case, FST and LD are both elevated across SD-Mal because of low recombination with other chromosomes from the same population, but high FST does not necessarily entail high LD and vice versa.

l.293 It seems a bit unlikely that both inversions arose at the same time. More likely, the distal one was segregating first, and then got hit by the proximal one. If this is a unique event, then you are simulating the second inversion origin, which is fine.

We agree. However, we do not know if they arose in an SD or wildtype background: the double inversion could also be segregating and become in cis with Sd by crossover, which would be a single event. We modified the text to clarify this point. “For our simulations, we assume that the acquisition of the second inversion (or the double inversion by crossover) was a unique event that enhanced drive strength and/or efficiency, and the onset of the selective sweep occurred following this event. ” (lines 318-319)

l.295 I wonder if one could use the Zeng et al. results on the SFS etc. to make inferences. A problem with what they've done is that they assume a static population size, rather than fitting a demographic model to the wild-type population; this would surely affect the skew in the SFS. Finally, silent sites should strictly be used for this analysis, as they should show less skew in the SFS than nonsynonymous sites.

Thank you for the suggestions. We repeated these analyses, using only SNPs located in noncoding regions (intronic and intergenic). We also included an exponential growth model based on demographic parameters from Kapopoulou et al. 2018. See Figure 4 and Suppl. Figure S6. Our conclusions do not change.

l.331 References about balancers are not really relevant to natural inversions.

While we agree that balancer chromosomes may be different because they are multiply inverted (although In(2R)Mal is not a simple inversion either) and not found in natural populations, we respectfully disagree that it is not relevant for our main point that inversions suppress recombination beyond their breakpoints. We added a citation here to Fuller et al. 2017, which studies natural inversions in D. pseudoobscura and elected to keep the other citations as well, but make the distinction (lines 360362): “First, chromosomal inversions can suppress recombination ~1-3 Mb beyond their breakpoints (in both multiply-inverted balancer chromosomes [Miller et al. 2016; Miller et al. 2018; Crown et al. 2018] and natural inversions [Stevison et al. 2011; Fuller et al. 2017]), extending the size of the sweep signal.”

l.348 I would be interested to see a more quantitative statement about the frequency of lethals in Sd-Mal. There are old investigations of inversions by Mukai's group, which suggested that normal inversions are enriched for lethals, but not by any means fixed: 1974 Genetics 76:339, 1976 82:63.

See Presgraves et al. 2009 and Brand et al. 2015 for a more extensive analysis on lethals in SD-Mal. They show that most combinations of two different SD Mal chromosomes are viable and fertile in complementation tests, suggesting that most of the deleterious mutations causing lethality or sterility are unique to each haplotype.

l.356-9 The sample size needs to be taken into account; it's presumably smaller for Sd-Mal than wildtype, and so Sd-Mal will get LD extending out much further purely by sampling. In the absence of LD, the expectation or r^2^ is 1/sample size, so will be substantial for a small sample. In any case, it's the product of Ne and recombination rate that determines r^2^, and Ne is certainly smaller for Sd-Mal. The recent sweep of course complicates interpretation of LD within the inversions. I think these points need attention.

The sample size for all population genomics analyses are comparable: 9 for SD-Mal, and 10 for each of the SD+ chromosome samples (In(2L)t and In(2R)NS). We have added sample sizes to the figure caption.

The main question we aimed to answer here was whether SD-Mal chromosomes have histories with or without recombination. The aim of Figure 5a is to show that LD decays with distance, a signature of past recombination. A secondary observation is that LD is higher for SD-Mal than wildtype. We agree that higher LD for SD-Mal than wildtype is expected, given the smaller Ne of SD-Mal chromosomes and, to a lesser extent, due to the smaller sample (9 versus 10). We acknowledge both factors in the manuscript (lines 397-398): “This pattern is not surprising: the low frequency of SD-Mal makes SDi/SDj genotypes, and hence the opportunity for recombination, rare. The smaller sample size of SD (n=9) versus SD+ (n=10) may also contribute to higher LD. “.

A meaningful quantitative comparison between SD and wildtype samples is not straightforward, however, as the answer depends on the finer details of the SD-Mal sweep. We are reluctant to go beyond our conclusions that a simple sweep model with a recent time of onset fits the data better than the standard neutral model.

l.385 Miller et al. 2016 Genetics 203:159 should also be cited. They get a similar mean tract length but a different rate of initiation from Comeron; the latter probably suffers from technical problems and is too high (it's much higher than the old Chovnick values).

Cited. Thanks.

l.398 These reference to deleterious effect of nonsynonymous mutations are rather old; some of the more recent stuff on estimating the DFE for them should probably be mentioned.

We added additional references.

l.400 There are problems with using N/S as a measure of deleterious effects, as the denominator is affected by Ne, and so N/S is automatically inflated if S is reduced by low Ne (see Campos et al. 2014 MBE 31:1010 for a discussion and a proposed alternative using the mean frequencies of segregating nonsyn mutations). In many ways, this is all a bit trivial, since, if Ne is reduced, the product Ne x s is automatically reduced, and this is what is used as a measure of selection.

The argument that N/S is “automatically inflated if S is reduced by low Ne” presumes that N mutations and S mutations have different distributions of Ne*s. Assuming the standard neutral model, N and S are assumed to be strictly neutral and linear with Ne, and any reduction in Ne should affect both the numerator and denominator of N/S equivalently. Instead, we find that N/S is >2-fold higher for SD versus SD^+^ chromosomes (Table 4). This cannot be explained by a simple reduction of Ne under the SNM and, therefore, implicates a difference in Ne*s between N and S mutations. As we argue in the manuscript, the fact that N/S is higher for SD than SD^+^ is as expected if N sites are enriched for slightly deleterious mutations, and if the efficacy of purifying selection is compromised on SD-Mal chromosomes.

We are reluctant to push these analyses further by trying to make strong inferences about Ne*s from the SFS, as suggested, for two reasons. First, our sample is rather shallow, comprising just 9 SD chromosomes. Second, the SD-Mal population is in a sweep recovery phase and not at any constant-size equilibrium (as typically assumed for low-recombination regions subject to recurrent hitchhiking effects; see, e.g., Campos et al. 2014).

Table 3 Again, there is an independence issue with the chis-squared test, due to shared geneaologies of the SNPs. I am no statistician, but the chi-squared test is likely to be too liberal.

The individual observations (mutations) in our samples are all affected by the nonindependence of genealogical history. But this is true for all contingency-based tests in population genetics, from the MK test to GWAS analyses. We see no immediate solution to this problem.The null hypothesis of this chi-squared test is whether the distribution of mutations into N and S classes is independent of whether they come from SD-Mal and SD+ samples. Our chi-square analyses strongly reject this null hypothesis. We agree it is important to be cautious when p values are marginal, but our Table 4 shows p-values smaller than 0.001, so we believe that our conclusions will remain valid.

l.434-433 This statement is too definite for my tastes. Supergenes are an observation; epistatic selection is an interpretation; there are other explanations for non-driving inversions, for example.

Thanks for the comment. We modified this statement (lines 475-477) to say: “Supergenes are balanced, multigenic polymorphisms. Under the classic model of supergene evolution, epistatic selection among component loci favors the recruitment of recombination modifiers that reinforce the linkage of beneficial allelic combinations.”

l.451-452 These properties are shared by non-driving inversions; it would be good to comment briefly on this commonality.

We added a brief comment on this (line 490-492): “Fourth, despite rare crossovers among complementing SD-Mal haplotypes and gene conversion from wildtype chromosomes, the relative genetic isolation and low frequency of SD-Mal results in the accumulation of deleterious mutations including, especially, TE insertions.“

l.455 The likelihood of significant degeneration must depend on the size of the non-recombining region; there is reason to be sceptical of claims about it for ones that involve a small regions.

We agree that the magnitude of Hill Robertson effects, and thus degeneration, are functions of the size of the region. However, here we are not discussing small regions. We hope this point is already clear in our pointing to Y chromosomes and supergene examples.

Reviewer #3 (Significance (Required)):This provides a significant advance in our knowledge and understanding of the Segregation Distorter complex of *Drosophila melanogaster*, the classic example of a selfish chromosome.

Thank you all for your constructive feedback.

References:

Charlesworth B, Hartl D L, 1978 Population dynamics of the segregation distorter polymorphism of *Drosophila melanogaster*. Genetics 89: 171–192.

Temin R G, Kreber R, 1981 A look at SD (Segregation Distorter) in the wild population in Madison, Wisconsin, more than 20 years after its initial discovery there. Drosoph. Inf. Serv. 56: 137.

Ghavi-Helm, Yad, et al. "Highly rearranged chromosomes reveal uncoupling between genome topology and gene expression." Nature genetics 51.8 (2019): 1272-1282.

Meadows, L. A., Chan, Y. S., Roote, J., and Russell, S. (2010). Neighbourhood continuity is not required for correct testis gene expression in *Drosophila*. PLoS biology, 8(11), e1000552.

Torosin, Nicole S., et al. 3D genome evolution and reorganization in the *Drosophila melanogaster* species group. PLoS genetics, 2020, vol. 16, no 12, p. E1009229.

Miller, D. E., K. R. Cook, A. V. Arvanitakis and R. S. Hawley, 2016 Third Chromosome Balancer Inversions Disrupt Protein-Coding Genes and Influence Distal Recombination Events in *Drosophila melanogaster*. G3 (Bethesda) 6: 1959-1967.